# Efficient Parallelized Simulation of Cyber-Physical Systems

**Bas van der Heijden**                                                      *b.heijden@hotmail.com*
*Cognitive Robotics*
*Delft University of Technology*

**Laura Ferranti**                                                           *l.ferranti@tudelft.nl*
*Cognitive Robotics*
*Delft University of Technology*

**Jens Kober**                                                               *j.kober@tudelft.nl*
*Cognitive Robotics*
*Delft University of Technology*

**Robert Babuska**                                                           *r.babuska@tudelft.nl*
*Cognitive Robotics, Delft University of Technology,*
*CIIRC, Czech Technical University in Prague*

**Reviewed on OpenReview:** *https://openreview.net/forum?id=VzKXbCzNoU*

## Abstract

Advancements in accelerated physics simulations have greatly reduced training times for reinforcement learning policies, yet the conventional step-by-step agent-simulator interaction undermines simulation accuracy. In the real world, interactions are asynchronous, with sensing, acting and processing happening simultaneously. Failing to capture this widens the sim2real gap and results in suboptimal real-world performance. In this paper, we address the challenges of simulating realistic asynchronicity and delays within parallelized simulations, crucial to bridging the sim2real gap in complex cyber-physical systems. Our approach efficiently parallelizes cyber-physical system simulations on accelerator hardware, including physics, sensors, actuators, processing components and their asynchronous interactions. We extend existing accelerated physics simulations with latency simulation capabilities by constructing a 'supergraph' that encodes all data dependencies across parallelized simulation steps, ensuring accurate simulation. By finding the smallest supergraph, we minimize redundant computation. We validate our approach on two real-world systems and perform an extensive ablation, demonstrating superior performance compared to baseline methods.

## 1 Introduction

Physics simulations on accelerator hardware (NVIDIA, 2020; Hu et al., 2020; Freeman et al., 2021; Todorov et al., 2012) have significantly reduced training times for reinforcement learning policies that conform to traditional, sequentially-structured agent-simulator interactions (Rudin et al., 2022). Such interactions lead to clear-cut and predictable execution paths, allowing for efficient parallelization, as shown in Fig. 1a. However, this sequential approach fails to capture the concurrent and dynamic nature of the real world.

Accounting for latency is crucial in the simulation of cyber-physical systems (CPS), which integrate computational algorithms with physical processes. In CPS, embedded computers and networks both monitor and control these processes, typically through feedback loops where physical processes impact computations and vice versa via sensors and actuators (Baheti & Gill, 2011). A critical application of CPS is vehicular platooning, involving multiple vehicles that operate in close proximity, coordinating their actions based on shared sensor data in real-time. This coordination is highly sensitive to time delays, making the accurate simulation of these delays critical for developing robust platooning algorithms (Zhao et al., 2021).

Delayed sensor data causes an agent to choose actions based on outdated information. Similarly, slow policy evaluation can unavoidably extend the effect of previous actions beyond their planned duration. Moreover, the focus has traditionally been on single agents trained end-to-end (Singh et al., 2019). In practice, however, AI systems deployed in real-world settings often rely on a pipeline of models. Accounting for the latency between these models will become crucial as tasks grow in complexity (Nishihara et al., 2017). Finally, physics simulators often bundle physics, sensor, and actuator simulations into a single unit running at a single rate. However, in reality, there are vital asynchronous effects within this block that need to be accounted for. Overlooking these asynchronous effects in simulation widens the sim2real gap and can lead to policies that do not perform well in the real world.

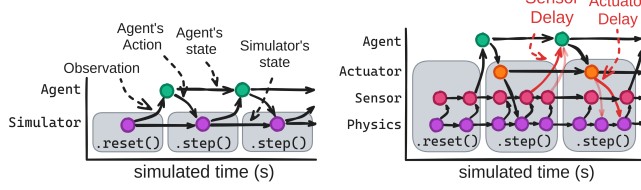

(a) Sequential interaction     (b) Asynchronous interaction

Figure 1: Comparative illustration of computation graphs with and without simulated delays. Vertices represent periodic computations, and edges represent data dependencies. (a) The absence of delay simulation creates consistent blocks of computation, enabling efficient parallelization across simulation steps yet failing to capture the inherent asynchrony of the real world. (b) While improving simulation fidelity, simulated delays between various components turn every simulation step into a diverse mix of computation, challenging parallelization efficiency.

To represent the asynchronous interactions between components, we advocate the division of the simulator into separate parts. This matches the typical design in robotics, where systems consist of interconnected nodes operating asynchronously at various rates (Quigley et al., 2009). This division enables the creation of computation graphs that accurately represent data flow in real-world situations, including latency effects. Consequently, each simulation step turns into a diverse mix of computation units from various components that run at different time scales, as illustrated in Fig. 1b. These must be executed in a sequence that respects the data dependencies outlined by the graph's edges. Simulating with these diverse partitions improves accuracy but complicates parallelization (i.e., simulating multiple copies of the simulation in parallel), as distinct partitions may need to execute simultaneously across GPU threads, hindering GPU efficiency. Such misalignment can happen with independent episodic resets, often initiated based on variable reset criteria. One parallel simulation might reset because the agent reached its goal, while another continues because the agent is still far away. Diverging execution paths can significantly reduce kernel efficiency (Shibata, 2010). When GPU threads take different paths, they must be serialized, leading to more instructions and reduced performance.

The main contribution of this paper is an approach to parallelize cyber-physical system simulations that emulates asynchronicity and delays with minimized computational overhead on accelerator hardware. This allows existing accelerated physics simulations to be extended with efficient latency simulation capabilities. We achieve this by identifying a graph—ideally the smallest one possible—that encodes all the data dependencies outlined by every simulation step's edges. This universal graph, referred to as a *supergraph*, is determined prior to simulation. Sorting the supergraph topologically yields a static execution order for parallel processing of simulation steps without violating data dependencies. By targeting the smallest supergraph, we minimize redundant computation. Finding the smallest supergraph is generally a complex, NP-hard problem (Trimble, 2023); however, our greedy algorithm efficiently approximates this supergraph by leveraging the inherent periodicity in cyber-physical systems.

In sum, we make four key claims: Our approach (i) emulates asynchronicity leading to more accurate simulation, (ii) efficiently handles time-scale differences and asynchronicity, resulting in higher parallelized simulation speeds than baseline approaches, and (iii) scales to complex system topologies. These claims are supported by an experimental evaluation on two real-world robotic systems, followed by a scalability analysis on two cyber-physical system topologies: vehicle-to-vehicle (V2V) platooning (Zhao et al., 2021) and unmanned aerial vehicle (UAV) swarm control (Asaamoning et al., 2021). An ablation study on the effects of the algorithmic simplifications was also conducted and included as an appendix. Finally, a motivational video for our approach is included as supplementary material, together with example source code to efficiently compile parallelized training environments with our approach.

## 2 Preliminaries

Before diving into the details of our approach, we first lay down some basic definitions and notation that will aid in the formalization of our problem and the description of our approach. We consider graphs $\mathcal{G} = (V, E)$ consisting of a set of vertices $V(\mathcal{G})$ and a set of directed edges $E(\mathcal{G})$. Edge $(u, v) \in E(\mathcal{G})$ denotes an edge from vertex $u$ to vertex $v$. The notation $|V(\mathcal{G})|$ denotes the number of vertices in $\mathcal{G}$. Any subset of vertices $V' \subseteq V(\mathcal{G})$ induces a unique *subgraph* $\mathcal{G}' \subseteq \mathcal{G}$. The *difference* $\mathcal{G}_2 - \mathcal{G}_1$, where $\mathcal{G}_1 \subseteq \mathcal{G}_2$, yields a graph $\mathcal{G}$ with $V(\mathcal{G}) = V(\mathcal{G}_2) \setminus V(\mathcal{G}_1)$ and $E(\mathcal{G}) = E(\mathcal{G}_2) \setminus E(\mathcal{G}_1)$. The edges that connect $\mathcal{G}_1$ and $\mathcal{G}_2 - \mathcal{G}_1$ are defined as the *cut-set* $C_{\mathcal{G}_2}(\mathcal{G}_1)$, which is a subset of $E(\mathcal{G}_2)$. The *union* of graphs $\mathcal{G}_1$ and $\mathcal{G}_2$ with respect to a set of edges $E_{12}$ is denoted as $\mathcal{G} = \mathcal{G}_1 \cup_{E_{12}} \mathcal{G}_2$, where $V(\mathcal{G}) = V(\mathcal{G}_1) \cup V(\mathcal{G}_2)$ and $E(\mathcal{G}) = E(\mathcal{G}_1) \cup E(\mathcal{G}_2) \cup E_{12}$. The *addition* $\mathcal{G}_1 + \mathcal{G}_2$, where $\mathcal{G}_1, \mathcal{G}_2 \subseteq \mathcal{G}_3$, yields a subgraph $\mathcal{G}_{12} \subseteq \mathcal{G}_3$, by unifying $\mathcal{G}_1 \cup_{E_{12}} \mathcal{G}_2$ where $E_{12} = C_{\mathcal{G}_3}(\mathcal{G}_1) \cap C_{\mathcal{G}_3}(\mathcal{G}_2)$. An *edge contraction* on an edge $(u, v) \in E(\mathcal{G})$ yields a new graph $\mathcal{G}'$ such that $V(\mathcal{G}') = V(\mathcal{G}) \setminus \{u, v\} \cup \{w\}$ and

$$E(\mathcal{G}') = (E(\mathcal{G}) \setminus \{(u, v), (v, u)\})$$
$$\cup \{(w, x) \mid (u, x) \in E(\mathcal{G}) \text{ or } (v, x) \in E(\mathcal{G})\}$$
$$\cup \{(x, w) \mid (x, u) \in E(\mathcal{G}) \text{ or } (x, v) \in E(\mathcal{G})\}.$$

The *ancestors* of a vertex $A_{\mathcal{G}}(u)$ are all vertices $V'(\mathcal{G}) \subseteq V(\mathcal{G})$ that can reach $u$ via a directed path in $\mathcal{G}$. The *roots* of a graph $\mathcal{G}$ are the set of vertices that have no incoming edges, formally $R(\mathcal{G}) = \{u \in V(\mathcal{G}) \mid \forall v \in V(\mathcal{G}), (v, u) \notin E(\mathcal{G})\}$. Similarly, the *leafs* of a graph $\mathcal{G}$ are the set of vertices that have no outgoing edges. A *Directed Acyclic Graph (DAG)* is a directed graph that contains no cycles. A *topological sort* $\tau$ of a directed acyclic graph $\mathcal{G}$ is a linear ordering of its vertices such that for every directed edge $(u, v) \in E(\mathcal{G})$, vertex $u$ comes before $v$ in the ordering. Multiple topological sorts may exist for a given graph $\mathcal{G}$, and the set of all possible topological sorts is denoted by $\mathcal{T}(\mathcal{G})$. A *labeling function* $L : V \to l$ is a function that assigns a label to each vertex. The set of all vertices with label $l$ is denoted by $V_l(\mathcal{G})$ and is arranged as a sorted list consistent with a topological sort of $\mathcal{G}$. We denote the set of topological sorts where the final vertex is of label $l$ in $\mathcal{G}$ as $\mathcal{T}_l^{-1}(\mathcal{G})$. Formally, this is defined as:

$$\mathcal{T}_l^{-1}(\mathcal{G}) = \{\tau \in \mathcal{T}(\mathcal{G}) \mid I(\tau, u) = |\mathcal{G}|, u \in V_l(\mathcal{G})\},$$

where $I(\tau, u)$ gives the position of vertex $u$ in the sorted set $\tau$. A *matching function* $f_{\mathrm{m}} : V \times V \to \{\text{True}, \text{False}\}$ is defined as follows:

$$f_{\mathrm{m}}(u, v) = \begin{cases} \text{True} & \text{if } L(u) = L(v), \\ \text{False} & \text{otherwise.} \end{cases}$$

A *mapping* between two graphs $\mathcal{G}_1$ and $\mathcal{G}_2$ is a bijective function $M : V'(\mathcal{G}_1) \to V'(\mathcal{G}_2)$ where $V'$ represent a subset of the vertices. Its domain $\text{dom}(M)$ is $V'(\mathcal{G}_1)$ and its range $\text{rng}(M)$ is $V'(\mathcal{G}_2)$. Operations like union $\cup$, intersection $\cap$, and difference $\setminus$ can be applied to both $\text{dom}(M)$ and $\text{rng}(M)$. A mapping $M$ can extend to $M'$ by adding a new vertex pair $(u, v)$ with $M' = M \cup \{(u, v)\}$ where $u \in V(\mathcal{G}_1) \setminus \text{dom}(M)$ and $v \in V(\mathcal{G}_2) \setminus \text{rng}(M)$. A *subgraph monomorphism* $M : V(\mathcal{G}_1) \to V'(\mathcal{G}_2)$ is a specialized mapping that maps each vertex $u$ to $v$ such that $L(u) = L(v)$ and each edge $(u, v)$ corresponds to an edge $(M(u), M(v))$ in $\mathcal{G}_2$. If such $M$ exists, $\mathcal{G}_2$ is a *supergraph* of $\mathcal{G}_1$ and can be reduced to $\mathcal{G}_1$ by removing vertices and edges in $\mathcal{G}_2$. The transformed set of edges $E_M(\mathcal{G}_1)$ under the mapping $M$ is defined as follows:

$$E_M(\mathcal{G}_1) = \{(u', v') \mid (u, v) \in E(\mathcal{G}_1),$$
$$u' = M(u) \text{ if } u \in \text{dom}(M), u' = u \text{ otherwise},$$
$$v' = M(v) \text{ if } v \in \text{dom}(M), v' = v \text{ otherwise}\}$$

This set includes edges $(u', v')$ where $u'$ and $v'$ are either mapped vertices of $u$ and $v$ under $M$ if they are in the domain of $M$, or are $u$ and $v$ themselves otherwise.

# 3  Our Approach

Consider the set of computation graphs generated by multiple episodes of an asynchronous system, as illustrated in Fig. 2, where vertices of the same color represent the same periodic computation unit, and edges represent data dependencies. These graphs might be partially recorded from real-world executions or synthetically created to reflect expected computation and communication delays. Variations in these graphs across episodes lead to distinct execution paths. However, managing these variations with conventional if-else branching for parallel execution on GPUs is inefficient, as highlighted in (Shibata, 2010). Predication (Taylor & Li, 2011) is a technique that sidesteps the need for if-else branching by executing all possible paths and masking out the computations that are not needed. This approach, while eliminating branching, can be inefficient due to the execution of all vertices in the paths, making it crucial to minimize the number of vertices.

To achieve this, we introduce an approach to identify a *minimum common supergraph* (mcs) that is acyclic and encapsulates all potential execution paths from a collection of computation graphs, optimizing for the fewest vertices (i.e., computational overhead).

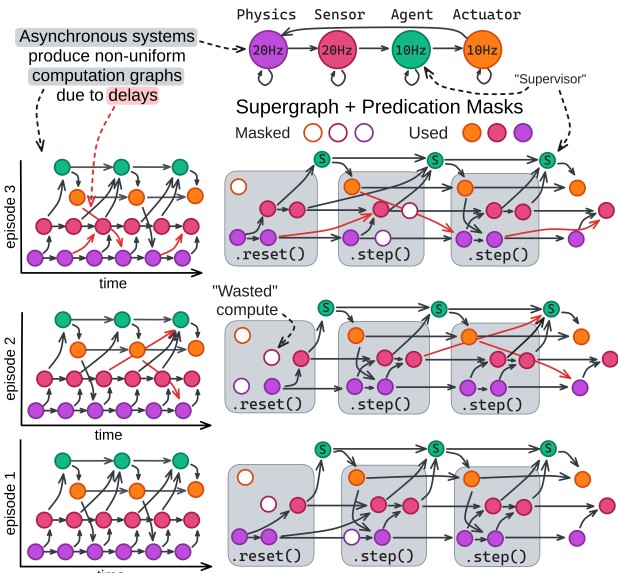

Figure 2: This figure illustrates our approach to efficiently simulating multi-rate asynchronous systems. Given variable delays, computation graphs can differ across episodes (left). We find a supergraph and predication masks, illustrated by the grey shaded blocks, for every computation graph that enables parallel execution across partitions (right). This mask, randomized during simulation, allows us to efficiently emulate asynchronicity and time-scale differences with minimal computational waste.

Topologically sorting the supergraph yields an execution order that, via predication masking, is transformed into a valid order for any given graph, as the supergraph encodes all data dependencies.

In aligning with standard simulator interfaces (Brockman et al., 2016), illustrated in Fig. 1, we first *partition* these computation graphs into disjoint subgraphs, each corresponding to a simulation step. Crucially, we designate a *supervisor* node in each partition, a pivotal element that dictates the boundaries of these subgraphs. In the context of reinforcement learning, the supervisor node is akin to the agent, while all other nodes within the partition form the environment, providing observations to and receiving actions from the supervisor node. The supervisor node's operating rate sets the simulation time step, ensuring that each partition accurately reflects a step of the simulation process. We then find a supergraph that accommodates all possible paths in every partition with a minimum number of vertices. This supergraph serves as a template that can be reduced to match any of the partitions (i.e., simulation steps) by masking (i.e., removing) specific vertices and edges. This setup enables parallel execution of any partition on accelerated hardware.

## 3.1  Problem Definition

Consider a set of observed computation graphs denoted by $\{\mathcal{G}_0, \mathcal{G}_1, \dots\}$, where each $\mathcal{G}_i$ is a DAG. For a given supervisor label $s$, our goal is to partition each $\mathcal{G}_i$ into disjoint subgraphs $\mathcal{P}_{i,1}, \mathcal{P}_{i,2}, \dots$. Each subgraph corresponds to a discrete simulation step and contains exactly one leaf vertex labeled as $s$. The objective is to determine these valid partitions along with the smallest DAG, $\mathcal{S}$, that serves as a common supergraph for all partitions. Similar to each partition, a single instance of the designated supervisor vertex in $\mathcal{S}$ must be a leaf vertex. Here, 'smallest' is defined by the number of vertices to minimize computational overhead. We aim to find a subgraph monomorphism $M_{i,j} : V(\mathcal{P}_{i,j}) \to V'(\mathcal{S})$ for each partition $\mathcal{P}_{i,j}$. This mapping allows us to reduce $\mathcal{S}$ into $\mathcal{P}_{i,j}$ using a predication mask. The predication mask is a binary mask applied to $\mathcal{S}$ to selectively remove vertices and edges not present in $\mathcal{P}_{i,j}$. Specifically, the mask is false for vertices and edges not in $\text{rng}(M_{i,j})$ and $E_{M_{i,j}}(\mathcal{P}_{i,j})$, respectively, and true otherwise.

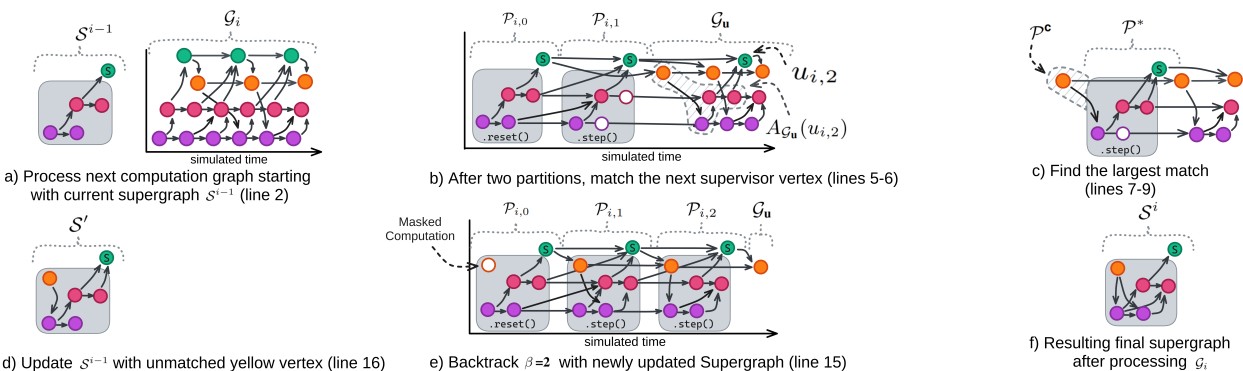

Figure 3: Illustration of the Minimum Common Supergraph Search process (Alg. 1) at a midway point. First, read from left to right, then top to bottom. A partial match is found for $u_{i,2}$, leading to an updated $\mathcal{S}'$ with missing ancestors $\mathcal{P}^-$ and initiating a backtrack to re-evaluate previous partitions.

## 3.2 Supergraph Search

**Algorithm 1:** Minimum Common Supergraph Search (mcs)

**Input:** Designated supervisor label $s$
**Input:** Number of steps to backtrack $\beta$
**Input:** A set of observed computation graphs $\{\mathcal{G}_0, \mathcal{G}_1, \dots\}$
**Output:** A set of partitions $\{\mathcal{P}_{0,1}, \mathcal{P}_{0,2}, \dots \mathcal{P}_{i,j} \dots\}$
**Output:** A supergraph $\mathcal{S}$ and mapping $M_{i,j}$ for all partitions $\mathcal{P}_{i,j}$

1   $\mathcal{S} \leftarrow$ Initialize with $V(\mathcal{S}) = \{u | L(u) = s\}$ and $E(\mathcal{S}) = \emptyset$
2   **for** $\mathcal{G}_i \in \{\mathcal{G}_0, \mathcal{G}_1, \dots\}$ **do**
3     $\mathcal{G}_u \leftarrow$ Initialize unmatched graph as $\mathcal{G}_i$
     /* Until all supervisor vertices are matched */
4     **while** $V_s(\mathcal{G}_u) \neq \emptyset$ **do**
5       $u_{i,j} \leftarrow$ Get next supervisor $u_i$ from sorted set $V_s(\mathcal{G}_u)$ with index $j = I(V_s(\mathcal{G}_i), u_i)$
6       $A_u \leftarrow u_{i,j}$ and its ancestors: $A_{\mathcal{G}_u}(u_{i,j}) \cup \{u_{i,j}\}$
7       $M^* \leftarrow$ Get largest map: Alg. 2 with $(s, \mathcal{S}, \mathcal{G}_u, A_u)$
8       $\mathcal{P}^* \leftarrow$ Partition subgraph: $\text{dom}(M^*) \subseteq V(\mathcal{G})$
9       $\mathcal{P}^- \leftarrow$ Missing subgraph: $A_{\mathcal{G}_u}(u_{i,j}) \setminus V(\mathcal{P}^*) \subseteq V(\mathcal{G})$
10      **if** $V(\mathcal{P}^-) = \emptyset$ **then**
       /* All ancestors were matched */
11        $M_{i,j} \leftarrow$ Store subgraph monomorphism $M^*$
12        $\mathcal{P}_{i,j} \leftarrow$ Store partition $\mathcal{P}^*$
13        $\mathcal{G}_u \leftarrow$ Remove matched partition: $\mathcal{G}_u - \mathcal{P}_{i,j}$
14      **else**
       /* Partial match */
15        $\mathcal{G}_u \leftarrow$ Restore $\beta$ partitions in $\mathcal{G}_u \subseteq \mathcal{G}_i$ with $\mathcal{G}_u + \mathcal{P}_{i,j-\beta} + \mathcal{P}_{i,j-\beta+1} + \cdots + \mathcal{P}_{i,j-1}$
16      $\mathcal{S} \leftarrow$ Update to $\mathcal{S}'$ with missing vertices and edges: $\bigl(V(\mathcal{S}) \cup V(\mathcal{P}^-), E(\mathcal{S}) \cup E_{M^*}(\mathcal{P}^* + \mathcal{P}^-)\bigr)$

Our approach, as outlined in Alg. 1 and illustrated in Fig. 3, aims to simultaneously achieve three main objectives: identifying the supergraph $\mathcal{S}$, determining the partitionings $\mathcal{P}_{i,j}$, and discovering the associated mappings $M_{i,j}$. For each computation graph $\mathcal{G}_i$, the algorithm iterates until all supervisor vertices are matched, as specified in Line 4. In every iteration, the largest partition $\mathcal{P}^*$ and its associated mapping $M^*$ are sought (Line 5-7), following the method detailed in Alg. 2 and explained later on in Sec. 3.3.

Depending on whether all ancestors are matched, the algorithm finds either a complete or a partial match corresponding to the supervisor vertex $u_{i,j}$. In the case of a complete match, both $\mathcal{P}^*$ and $M^*$ are stored (Line 10-13). For partial matches, the algorithm backtracks $\beta$ iterations to reconsider previously matched partitions (Line 15). In either case, the supergraph $\mathcal{S}$ is updated using Eq. (1) to ensure it remains a supergraph of its previous version and incorporates all necessary ancestors $V(\mathcal{P}^-)$ for future matches, as follows:

$$\mathcal{S}' = \bigl(V(\mathcal{S}) \cup V(\mathcal{P}^-), E(\mathcal{S}) \cup E_{M^*}(\mathcal{P}^* + \mathcal{P}^-)\bigr), \tag{1}$$

where $\mathcal{P}^* + \mathcal{P}^- \subseteq \mathcal{G}_i$. More edges in the updated supergraph $\mathcal{S}'$ effectively constrain the number of possible mappings for subsequent partitions by reducing the number of topological sorts available in the supergraph. Conversely, more vertices in $\mathcal{S}'$ increase its expressiveness by increasing the number of vertices that can be mapped to a vertex in subsequent partitions, but also increase the computational overhead of the simulation. In the next section, we detail the algorithm for finding the largest match, Alg. 2, which is a critical component of the supergraph search algorithm. It may only result in mappings $M^*$ that ensure that collectively contracting all edges $(u, v) \in E(\mathcal{P}^*)$ in $\mathcal{G}_u$ will not introduce any cycle in the resulting graph $\mathcal{G}'_u$. This constraint is critical to ensure that the updated supergraph $\mathcal{S}'$, as governed by equation Eq. (1), remains acyclic. This sets us apart from previous work (Bunke et al., 2000; 2003), which do not consider this constraint.

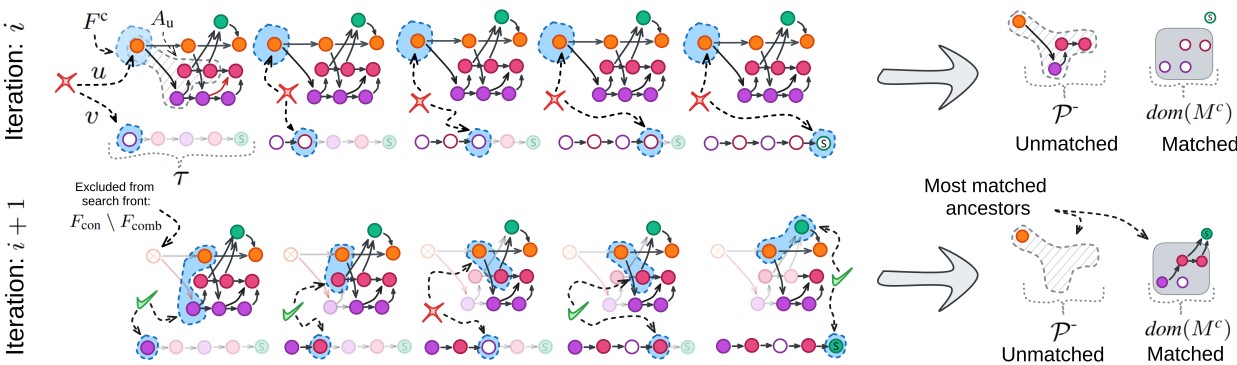

Figure 4: Midway illustration of Line 9-16 in the Largest Match Search (Alg. 2) linked to step (c) in Fig. 3. The ✓ and ✗ symbols indicate whether a vertex in the candidate front $F^c$ is matched in Line 12. The first iteration yields an empty mapping due to the absence of a yellow vertex in the supergraph. In the second iteration, the vertex is excluded from the front, enabling a large partial match. The search is only displayed for a single topological sort.

## 3.3 Largest Match Search

**Algorithm 2:** Largest Match Search

**Input:** Designated supervisor label $s$
**Input:** Supergraph $\mathcal{S}$
**Input:** Unmatched computation graph $\mathcal{G}_u$
**Input:** Vertices to be matched $A_u$
**Output:** Largest mapping $M^*$

1   $M^* \leftarrow$ Initialize an empty mapping
2   $\mathcal{G}_{\text{excl}} \leftarrow$ Initialize search graph as $\mathcal{G}_u$
3   **while** *True* **do**
4      $F_{\text{excl}} \leftarrow$ Initialize search front as roots $R(\mathcal{G}_{\text{excl}})$
5      $F_{\text{con}} \leftarrow$ Determine constrained front: $F_{\text{excl}} \cap A_u$
6      **forall** $F_{com} \in k\text{-}comb(F_{con})$ **do** // Greedy: one $F_{\text{com}}$ per $k$
7          **forall** $\tau \in \mathcal{T}_s^{-1}(\mathcal{S})$ **do** // Greedy: only a single $\tau$
8              $\mathcal{G}^c \leftarrow$ Remove $u \in F_{\text{con}} \setminus F_{\text{com}}$ from $V(\mathcal{G}_{\text{excl}})$
9              $F^c \leftarrow$ Initialize front: $F_{\text{excl}} \setminus (F_{\text{con}} \setminus F_{\text{com}})$
10             $M^c \leftarrow$ Initialize an empty candidate mapping
11             **forall** $v \in \tau$ **do**
12                 **if** $\exists u \in F^c : f_m(u,v) = True$ **then**
13                     $u \leftarrow \{u \in F^c : f_m(u,v) = \text{True}\}$
14                     $M^c \leftarrow$ Extend mapping: $M^c \cup \{(u,v)\}$
15                     $\mathcal{G}^c \leftarrow$ Remove matched $u$ from $V(\mathcal{G}^c)$
16                     $F^c \leftarrow$ Update front: $F^c \setminus \{u\} \cup R(\mathcal{G}^c)$
17             **if** $|dom(M^c) \cap A_u| > |dom(M^*) \cap A_u|$ **then**
18                 $M^* \leftarrow M^c$         /* Store largest mapping */
19             $s_{\text{max}} \leftarrow |A_u| - |A_u \setminus V(\mathcal{G}_{\text{excl}})| - (|F_{\text{con}}| - |F_{\text{com}}|)$
20             **if** $(|dom(M^*) \cap A_u|) \geq s_{max}$ *or* $|dom(M^*)| = |\mathcal{S}|$
                **then** **return** $M^*$
21      $\mathcal{G}_{\text{excl}} \leftarrow$ Exclude vertices from search graph:
       $V(\mathcal{G}_{\text{excl}}) \setminus F_{\text{con}}$

Our approach to identifying the largest valid mapping $M^*$ for each supervisor vertex $u_{i,j}$ in $\mathcal{S}$ is outlined in Alg. 2 and illustrated in Fig. 4. Initially, it tries to match all ancestor vertices in candidate subgraphs $\mathcal{G}^c$, extracted from the unmatched graph $\mathcal{G}_{\text{excl}}$ (Line 2). The search is refined by iteratively excluding ancestors in a breadth-first manner from $\mathcal{G}^c$ (Line 8, and 21).

To this end, a refined search front $F^c$ is formed for each iteration (Line 4-6, Line 9). The algorithm explores all $k$-combinations of $F_{\text{con}}$ in descending order of $k$ (Line 6). For each combination, the largest candidate mapping $M^c$ is sought by traversing all valid topologies of $\mathcal{S}$, starting from the refined search front $F^c$ (Line 7-10). Every time a match is found, the mapping is extended and the search front is updated (Line 11-16). Our approach assumes that vertices are stateful, i.e., vertices of similar labels are connected with one another, ensuring the uniqueness of the match (Line 12-13). If a larger mapping is found, $M^*$ is updated (Line 17-18). The algorithm halts the search if no larger match can be found (Line 19-20). To lower computational complexity, we apply crucial greedy approximations in lines 6 and 7, as further motivated in Sec. 3.4.

We may only consider mappings $M^*$ that ensure that the updated supergraph $\mathcal{S}'$ remains acyclic after updating with Eq. (1). To ensure this, we initiate each candidate search at the roots of $\mathcal{G}_{\text{excl}}$ and $\mathcal{S}$, as specified in Line 4 and 7, adopting a search strategy aligned with the topological sort of $\mathcal{S}$ and a breadth-first search of $\mathcal{G}_{\text{excl}}$. This approach guarantees that edges between matched vertices in $dom(M^*)$, represented by $\mathcal{P}^*$, cannot create cycles in $\mathcal{S}'$. For vertices not matched in $dom(M^*)$ (designated as $\mathcal{P}^-$), their positioning is either strictly prior to or following $\mathcal{P}^*$ in the topological sense, thus also ensuring acyclicity in $\mathcal{S}'$. This strict placement is achieved by initially removing only root vertices from consideration (Line 4) and subsequently

extending the search frontier solely upon removing a newly found match that subsequently leads to new root vertices (Line 15-16).

### 3.4 Limitations and Approximations

The efficacy of our approach is contingent on a set of assumptions. Firstly, the best performance is achieved when the computation graphs exhibit a recurring topological structure. Secondly, the model assumes substantial time-scale differences between what we term the *supervisor vertex* and other vertices. Finally, our approach assumes that vertices are stateful, i.e., vertices of similar labels are connected with one another. These assumptions are particularly well-suited for cyber-physical systems where components are stateful and run at fixed target frequencies, and where the supervisor vertex often takes the form of a slower, learning agent or an outer-loop controller. Moreover, the algorithm assumes that the computation graphs' structure does not depend on the data processed by the vertices. Specifically, we assume delays in the system are not a function of the internal states, outputs, or incoming inputs.

Identifying the minimal common supergraph is an NP-hard problem (Trimble, 2023). To manage this complexity, we make several approximations to Alg. 2. If all vertices are assumed to be stateful, then the constrained front $F_{\text{con}}$ can contain at most one vertex for each label, i.e. $|F_{\text{con}}| = |\text{rng}(L)|$. Then, the worst-case time complexity for considering all topological sorts of the supergraph $\mathcal{S}$ and all combinations of $F_{\text{con}}$ is $\mathcal{O}(2^{|\text{rng}(L)|} + |V|!)$ (Line 6-7 in Alg. 2). We alleviate this by considering only a single topological sort of $\mathcal{S}$ and a single combination per combination size $k$, reducing the worst-case time complexity to $\mathcal{O}(|\text{rng}(L) + 1| + |V|)$. We have found that these approximations do not significantly impact the resultant supergraph in our evaluations, as detailed in the ablation study in Appendix B. Lastly, the sequence in which computation graphs are processed can affect the resultant supergraph. Similar to Bunke et al. (2003), this has not proven to have a significant impact in our evaluations.

## 4 Experimental Evaluation

The main focus of this work is an efficient approach to simulate delays in parallelized simulation on accelerator hardware. We present our experiments to show the capabilities of our approach and to support our key claims that our approach (i) emulates asynchronicity leading to more accurate simulation, (ii) efficiently handles time-scale differences and asynchronicity, resulting in higher parallelized simulation speeds than baseline approaches, (iii) scales to complex system topologies. In the remainder of this section, we will use *mcs* to refer to our proposed method.

### 4.1 Baselines

We outline three baseline methods for our experimental evaluation. The *sequential* baseline (*seq*) assumes no delays in computation graph processing, illustrating a conventional approach as shown in Fig. 1a. This baseline serves as a reference for evaluating the impact of realistic delays in simulations.

We then introduce two baselines that incorporate delays by randomizing predication masks in parallelized simulations, but differ in supergraph construction. Given the absence of existing methods that can handle the DAG constraint and partitioning requirements for our supergraph (as discussed in Sec. 3.3), these baselines represent straightforward strategies for supergraph construction. Both baselines sequentially stack $K$ layers in the supergraph, with each layer containing a vertex for every non-supervisor label and concluding with a final layer of a single supervisor vertex. This structure ensures the supergraph is a DAG and with its size as $|\mathcal{S}| = K \times (|\text{rng}(L)| - 1) + 1$, thereby ensuring subgraph monomorphisms across partitions with an adequate number of layers. The *topological* baseline (*top*) sets $K$ equal to the number of vertices in the largest partition. While this method guarantees a subgraph monomorphism with each partition, it can lead to disproportionately large supergraphs with sparse layer utilization. The *generational* baseline (*gen*), on the other hand, sets $K$ as the maximum path distance across partitions. This approach is more space-efficient but also tends to over-include vertices, as it does not account for time-scale differences between vertices. Consequently, each layer incorporates every vertex label, even those infrequently used.

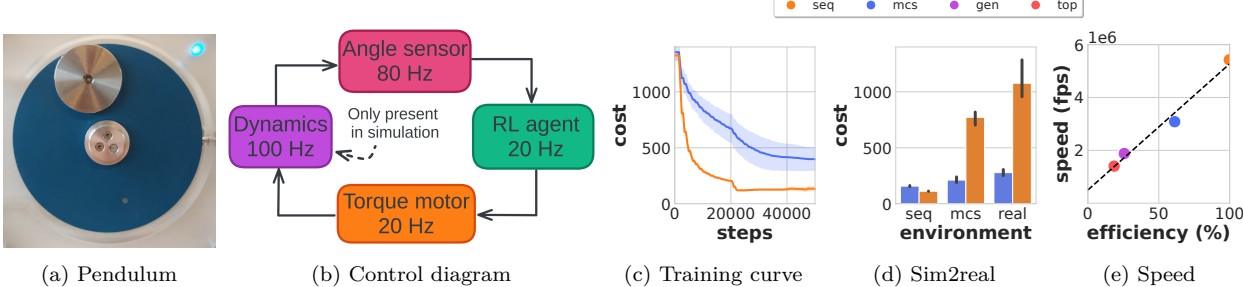

|   (a) Pendulum   |   (b) Control diagram   |   (c) Training curve   |   (d) Sim2real   |   (e) Speed   |

Figure 5: Sim2real evaluation of an RL policy trained to swing up a pendulum with (*mcs*) and without delays (*seq*). Panels (a) and (b) depict the experimental setup and control diagram, respectively. Panels (c) and (d) show that *mcs* outperforms *seq* in the real-world evaluation despite a slower convergence and lower simulated performance. This discrepancy is attributed to *seq*'s inability to develop a robust policy, as evidenced by cross-evaluations in panel (d) of *mcs* and *seq* in each others training environment, where *mcs* proved effective in both environments, unlike *seq*. Panel (e) shows that *mcs* achieves a higher mean simulation speed with a compacter supergraph (*mcs*: $|\mathcal{S}| = 13$) than baseline approaches that also consider delays (*gen*: $|\mathcal{S}| = 31$ and *top*: $|\mathcal{S}| = 43$).

To evaluate these methods, we introduce the *supergraph efficiency* metric ($\eta$):

$$\eta = 100 \times \frac{1}{N} \sum_{i,j} \frac{|\mathcal{P}_{i,j}|}{|\mathcal{S}|}$$

Here, $N$ denotes the total number of partitions, with $\eta$ indicating the mean partition size relative to the total supergraph size. This metric effectively quantifies the proportion of vertices actively utilized (unmasked) in emulating the computation graphs across episodes. Note that a 100% efficiency may not be achievable in practice, as it would imply that all partitions have an equal number of vertices.

## 4.2 Performance

In this set of experiments, we aim to validate that randomizing predication masks during training enhances the fidelity of robotic simulations and our approach to identifying the supergraph leads to more efficient parallelized simulations. We validate the performance on two real-world systems: a pendulum swing-up task and a vision-based robotic manipulation task. We use two different control strategies, reinforcement learning (RL) and model predictive control (MPC), to demonstrate the utility of our approach in different real-world settings.

### 4.2.1 Pendulum swing-up task

The pendulum swing-up task is a well-known RL benchmark with nonlinear, unstable, and underactuated dynamics sensitive to delays (Derner et al., 2020). The choice for this task is deliberate; it highlights the core challenge of delay compensation in reinforcement learning. By demonstrating how neglecting delay simulation can impair policy transfer even in seemingly simple scenarios, we underscore the greater consequences for complex systems where delays are unavoidable and complexity is greater, as discussed in prior work (Liu et al., 2019; Asaamoning et al., 2021; Lou et al., 2019; Peters et al., 2014). The simplicity of the task serves to clarify the fundamental importance of accounting for delays in sim2real approaches.

The experimental setup and control diagram are depicted in Figures 5a and 5b, respectively. A failure to emulate the asynchronous real-world interactions between components makes a simulation-trained policy ineffective when transferred to a real-world setting. Policies were trained using soft actor-critic (SAC) (Haarnoja et al., 2018) in two simulators: one emulating delays (our approach: *mcs*) and another without delays (sequential approach: *seq*). Note that the *gen* and *top* baselines are not included in the sim2real evaluation. This exclusion is due to their replication of the same effective computation graphs as *mcs*, leading to identical policy outcomes. Hence, we only consider these baselines later on in the simulation speed evaluation within this section. We record 10 computation graphs from the real-world system to identify a

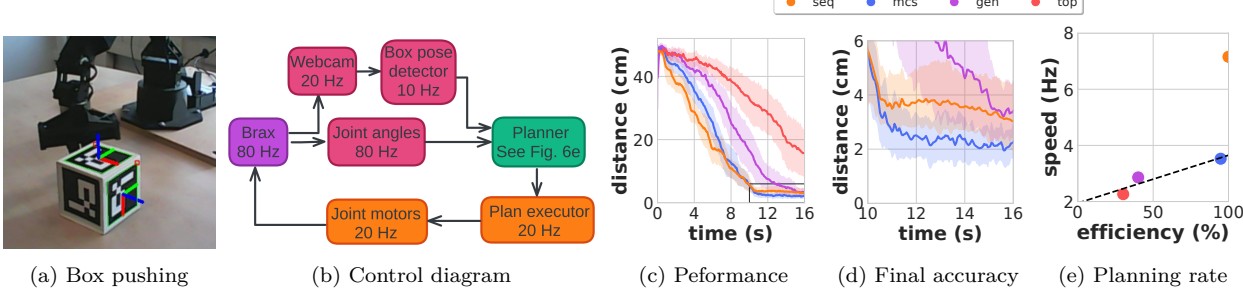

(a) Box pushing  (b) Control diagram  (c) Peformance  (d) Final accuracy  (e) Planning rate

Figure 6: A comparison of four MPC strategies for a task where a manipulator moves a box to a target: three consider delays (*mcs*, *gen*, *top*) and one does not (*seq*). Panels (a) and (b) depict the experimental setup and control diagram, respectively. Panels (c) and (d) depict the mean convergence rate and final accuracy over 10 episodes with 95% confidence intervals, respectively, while (e) correlates these with the achieved replanning rate. The *seq* strategy, although faster initially, leads to less accurate movements due to ignoring delays. The *mcs* method, while replanning less frequently, achieves approximately 40% higher accuracy. Moreover, *mcs* exhibits the highest replanning rate with a smaller supergraph (*mcs*: $|\mathcal{G}| = 54$) compared to *gen* ($|\mathcal{G}| = 139$) and *top* ($|\mathcal{G}| = 223$) that also consider delays.

supergraph, partitioning and corresponding predication masks that were randomized during training. Each experiment was replicated five times with different random seeds and the results are presented in Fig. 5. Though the sequential (*seq*) approach exhibits quicker convergence and superior simulated performance, it underperforms in real-world tests compared to our approach that includes latency simulation during training. A smaller performance gap between simulation and reality suggests that our approach leads to more accurate simulation, yielding more effective real-world policies. This is further supported by cross-evaluations of the trained policies in each other's training environment, where *mcs* proved effective in both environments, unlike *seq*.

On average, it took 0.54 seconds to identify the supergraph and predication masks for the 10 recorded computation graphs, which is a one-time startup cost that is small compared to the total training and compilation time of 100 seconds. To establish the link between efficiency and simulation speed, we carried out a parallelized performance evaluation of the swing up-task on an RTX 3070 GPU. We deliberately measure simulation speed during policy evaluation rather than measuring the overall training time to clearly separate simulation speed improvements from any learning algorithm and training-related overhead. We compiled the supergraph with JAX (Frostig et al., 2018) and randomized the predication masks across 1000 parallelized episodes. We used the supergraphs produced by our approach with backtracking $\beta = 5$ and both baseline methods and recorded the simulation frames per second (fps). As indicated in Fig. 5, our method notably outperforms other baselines that include delays, achieving an approximate simulation speed of 3 million fps. This improvement is largely attributed to a more compact supergraph. We observed a clear linear relationship between $\eta$ and simulation fps, which is consistent with the inverse proportionality between simulation fps and supergraph size.

### 4.2.2 Manipulation task

In the manipulation task, a Viper 300x robotic manipulator moves a box to a target based on streaming webcam images. The goal is to minimize the distance between the box and a goal position. Our experimental setup and control diagram are shown in Figures 6a and 6b. Emphasizing the importance of delay simulation, we use a consumer-grade Logitech C170 webcam, chosen for its low resolution, modest frame rate, and high latency, to track the box's position and orientation.

We adopt the MPC approach from Yang et al. (2020), planning actions based on the most recent robot observations using the Cross Entropy Method (CEM) (Rubinstein & Kroese, 2004). CEM, known for its efficient, derivative-free optimization, is particularly advantageous due to its parallelizability. Considering the contact-rich nature of box pushing, we opt for Brax (Freeman et al., 2021) as our dynamics model within the MPC framework, instead of learning complex contact dynamics. Brax, a differentiable physics simulator, is optimized for GPU acceleration and effectively handles contact-rich tasks. In a similar approach, Pezzato

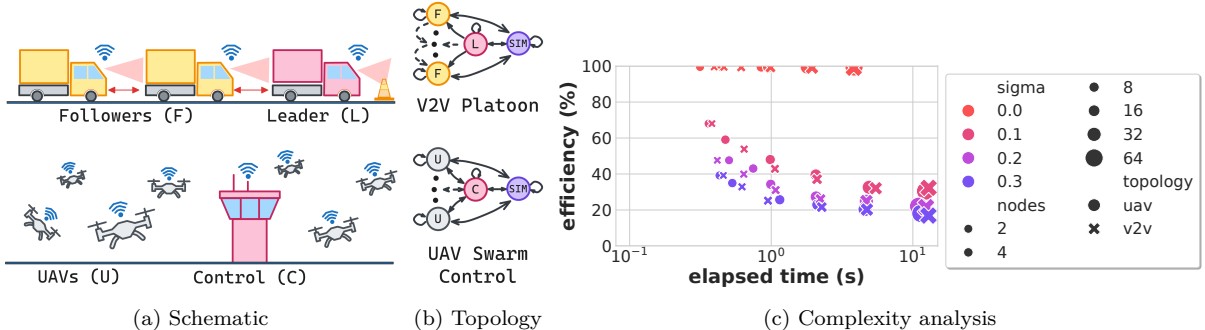

(a) Schematic          (b) Topology          (c) Complexity analysis

Figure 7: Panel (a) shows the V2V platooning and UAV swarm control systems, with the former comprising a leader and followers, and the latter a central controller and UAVs. Panel (b) depicts their respective topologies, where every component communicates at 20 Hz with each other, while the simulator runs at 200 Hz. The leader and controller are chosen as the supervisor nodes, respectively. Panel (c) shows the computational complexity versus efficiency for different topologies, asynchronicity levels, and node counts, highlighting their impact on performance.

et al. (2023) recently used PhysX (NVIDIA, 2020) to solve a box-pushing task. Our implementation employs CEM for three iterations, involving 75 samples per iteration and a planning horizon of two control steps, each lasting 0.15 seconds. We implement our approach using JAX (Frostig et al., 2018) and execute it on an RTX 3070 GPU.

We evaluate four MPC strategies: three accounting for delays (*mcs*, *gen*, *top*) and one ignoring them (*seq*). Delay-inclusive strategies, following Yang et al. (2020), use past plans to predict future box positions and orientations at action time. This prediction is based on the 10 recorded computation graphs of the system that are used to identify a supergraph, partitioning, and corresponding predication masks. On average, it took 1.53 seconds to identify the supergraph and predication masks for the 10 recorded computation graphs, which is a one-time startup cost that is small compared to the total evaluation time of 160 seconds. Due to their computational load, these strategies have a lower replanning rate compared to the delay-agnostic *seq*. The slower the replanning rate, the further into the future the planner must predict, increasing the likelihood of inaccurate predictions. As Fig. 6d shows, *mcs* achieves 40% higher accuracy than *seq*, despite less frequent replanning. Moreover, the *mcs* method also results in smoother operations than *seq*, as can be observed in the supplementary material's video. The larger supergraphs in *gen* and *top* result in excessively slow replanning, significantly reducing convergence rates, and final accuracy. This illustrates the trade-off between accuracy and efficiency, where the improved accuracy must justify the additional computational load.

## 4.3 Scalability

The next set of experiments support the claim that our approach scales to complex system topologies. In Sec. 4.2, we showed that employing a supergraph with randomized predication masks can effectively emulate direct delay simulation. We also identified an approximate linear correlation between graph efficiency $\eta$ and simulation speed. Next, we assess our method's scalability, analyzing various system topologies and modifying node counts and asynchronicity degrees to ascertain their effects on identifying efficient supergraphs. In this section, we consider two cyber-physical systems for which delay simulation is crucial: vehicle-to-vehicle (V2V) platooning (Zhao et al., 2021; Jia et al., 2015; Peters et al., 2014) and unmanned aerial vehicle (UAV) swarm control (Asaamoning et al., 2021). Furthermore, a detailed analysis of the impact of different abstract topological characteristics on supergraph efficiency is provided in Appendix A.

Fig. 7a illustrates the V2V platooning and UAV swarm control systems. In V2V platooning, vehicles maintain a set distance and speed, following a leader. This requires each vehicle to respond to the leader, highlighting the necessity for delay-aware simulation. Vehicles communicate with the leader and the vehicle ahead. For UAV swarm control, a central entity directs the UAVs to prevent collisions and achieve formation, with UAVs communicating solely with this controller. Additionally, each component connects to the simulator to

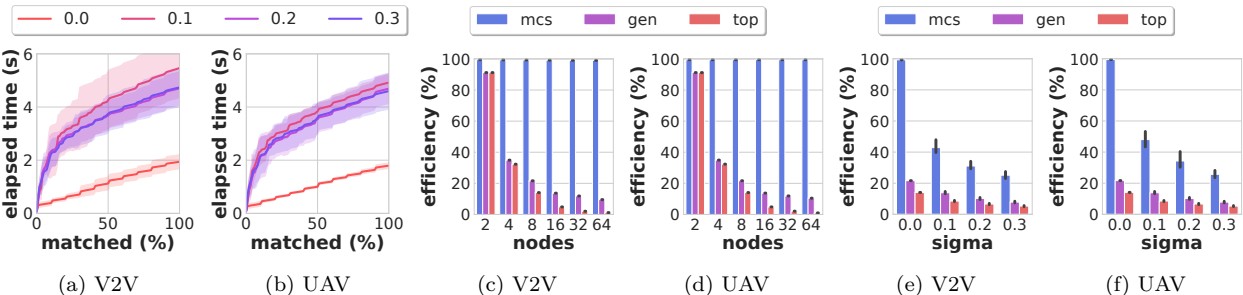

Figure 8: Panels (a) and (b) show the elapsed time for completion with $N = 32$ nodes across various asynchronicity. Initial episodes are time-intensive due to numerous preliminary partial matches, followed by a consistent linear time scaling in processing time. Panels (c-f) compare the efficiency of *mcs* (our approach), *top*, and *gen*. In panels (c) and (d) the number of nodes is varied with no asynchronicity ($\sigma = 0$), while in panels (e) and (f) the asynchronicity levels are varied with a fixed size of $N = 8$ nodes.

enable physics simulation. Accurate simulations require delays simulation in both systems, as discussed in Zhao et al. (2021) and Asaamoning et al. (2021). Within systems encompassing $N$ nodes, there is a single simulator and one leader or controller designated as the *supervisor*, alongside $N - 2$ Followers and UAVs, respectively, as illustrated in Fig. 7b. The simulator runs at 200Hz, while all other nodes communicate with each other at a target rate of 20Hz. The effective sampling time of every node $i$ is computed as $\Delta t_{i,k} = \Delta t_i + \max(0, x_k \Delta t_i)$, where $x_{i,k}$ is the delay of node $i$, experience during sequence number $k$, scaled with the node's nominal sampling time. An Ornstein-Uhlenbeck (OU) process (Bibbona et al., 2008) is used to model every node's delay to reflect the temporal correlation of delays, defined as follows:

$$x_k = \theta x_{k-1} + \sigma \nu, \tag{2}$$

where $\theta$ is a correlation coefficient, $\sigma$ is the standard deviation, and $\nu$ is a Gaussian random variable with zero mean and unit variance. The standard deviation of an OU process is related to the standard deviation of a Gaussian distribution with $\sigma_g = \sqrt{\frac{\sigma^2}{2\theta}}$. We artificially generate computation graphs for the topologies depicted in Fig. 7b, varying the asynchronicity level $\sigma \in \{0, 0.1, 0.2, 0.3\}$ and the number of nodes $N \in \{2, 4, 8, 16, 32, 64\}$. We replicate each experiment 5 times using different random seeds. For each configuration, we generate 10 computation graphs, each running for a duration of 10 seconds. Example computation graphs are presented in Appendix C. We employ Alg. 1 to identify a supergraph on a single core of an Intel Core i9-10980HK and compare its performance with two baseline approaches.

Fig. 7c presents an analysis of our method's computational complexity in constructing the supergraph, considering both the computation graph's characteristics $(N, \sigma)$ and topology (*v2v*, *uav*). We observe that efficiency is inversely related to the asynchronicity level and, to a lesser extent, to the number of nodes. Moreover, a decrease in efficiency correlates with an increase in computation time, primarily because fewer complete matches are found. Nevertheless, the one-time upfront cost of identifying the supergraph is usually minor when compared to the overall simulation time, substantiating our claim that our approach scales effectively to complex system topologies. Figures 8a and 8b detail the required computation time under varying levels of asynchronicity, as it processes all recorded computation graphs. The initial episodes incur higher computational costs due to the increased computational overhead of handling numerous partial matches (Line 6-7 in Alg. 2), while subsequent episodes demonstrate linear scaling in time.

Figures 8c and 8d compare the performance of our algorithm with baseline approaches for different numbers of nodes when there is no asynchronicity ($\sigma = 0$). Our approach achieves a 100% efficiency, whereas the efficiency of baseline approaches declines rapidly as the number of nodes increases. Figures 8e and 8f demonstrate the performance of our algorithm compared with baseline approaches for different levels of asynchronicity when the topologies comprise $N = 8$ nodes. As asynchronicity increases, partitions become more dissimilar, and the efficiency of our approach does decline, yet it remains multiples higher than that of the baseline approaches. The supergraphs generated by our method, along with those from the baseline approaches, are depicted in Appendix C.

In summary, our evaluation suggests that our method successfully emulates asynchronicity, offering more accurate and faster parallelized simulations compared to baseline approaches. At the same time, our method scales well to larger system topologies by finding more efficient supergraphs than baseline methods. Thus, we have substantiated all our key claims through this experimental evaluation.

## 5 Related Work

**Accelerated Physics Simulation**  Accelerated physics simulators like Brax (Freeman et al., 2021), MJX (Todorov et al., 2012), and PhysX (NVIDIA, 2020) are designed for GPU execution. However, they lack features for simulating delays between their physics engine and other components, such as sensors and actuators. Moreover, to mimic complete systems, these simulators must be extended with controllers and perception modules. Yet, these extensions typically interact with simulators sequentially, ignoring the concurrent and asynchronous nature of real-world systems. Our approach builds on this by dividing these simulators into separate components, facilitating the simulation of asynchronous interactions between them.

**Adressing Asynchronicity and Delays**  The ORBIT framework (Mittal et al., 2023) and research by Bouteiller et al. (2021) have explored integrating delays into robotic simulations. While ORBIT introduces actuator delays to PhysX, it overlooks the asynchronicity between other system components. Bouteiller et al. (2021)'s work centers on compensating for system delays in the learning algorithm, not addressing the dynamic interactions among delayed components. In contrast, our method extends beyond actuator delays, encompassing asynchrony across all components.

**Minimum Common Supergraph**  Our approach addresses a variant of the *minimal universal supergraph* (MUG) problem, which seeks the smallest supergraph, i.e., the mcs, containing all graphs in a given set as a subgraph (Trimble, 2021). Unlike the brute-force exact algorithm presented in Trimble (2021), which is suitable only for small graph sets, our approximate greedy algorithm is capable of handling graphs with more than 2000 vertices. In Bunke et al. (2003), an iterative update strategy, based on Bunke et al. (2000), is utilized to approximate the mcs. Our method shares similarities but satisfies an additional constraint: the resulting mcs must remain acyclic post-merge. Furthermore, our extended objective is to efficiently partition a provided set of larger graphs into smaller subgraphs before finding the mcs for these partitioned subgraphs. In contrast, Trimble (2021) and Bunke et al. (2003) start from a given and static collection of graphs and focus strictly on the identification of the mcs, meaning the partitioning we perform together with the supergraph identification is already a given in their scenario. Both Trimble (2021) and Bunke et al. (2003), and our method, (approximately) solve the *maximum common subgraph* problem as a subroutine to find the *minimum common supergraph* (mcs) (McCreesh et al., 2017; McGregor, 1982). However, our focus is on subgraph monomorphisms, which allow for additional edges in the subgraph, rather than induced subgraph isomorphisms, which require a one-to-one correspondence between every node and edge in the subgraph and target graph. To efficiently identify the largest mapping, we introduce an algorithm that leverages the acyclic nature of our mcs that accelerates the search for a large approximate mapping. Note that our algorithm restricts the largest mapping to connected subgraphs, potentially overlooking larger disconnected mapping candidates.

## 6 Conclusion

In this paper, we introduced a method for efficiently simulating inherently asynchronous systems on accelerator hardware. Our approach leverages recorded computation graphs from real-world operations to accurately model asynchronicity and time-scale differences. The experiments suggest that our approach provides a scalable, efficient, and accurate means for simulating cyber-physical systems. We evaluated our method in two real-world scenarios against baselines and confirmed its efficacy in emulating asynchronicity and handling time-scale differences efficiently. Our work opens avenues for developing fast and accurate cyber-physical system simulations. Finally, our approach holds promise for enhancing the integration of other machine learning algorithms that generate dynamic graphs into frameworks like Jax (Frostig et al., 2018), by aligning dynamic computation graphs with static ones.

## 7    Acknowledgments

This work is funded by the EU's H2020 OpenDR project (grant No 871449) and the Dutch Science Foundation NWO-TTW's Veni project HARMONIA (18165).

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

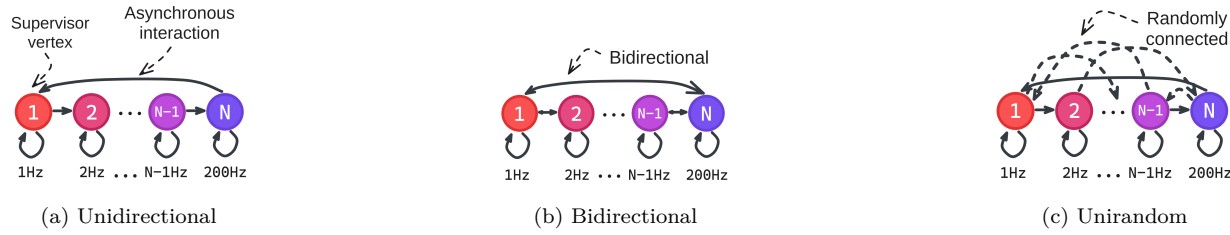

Figure 9: Three abstract topologies to evaluate the scalability of our approach. (a) *Unidirectional*: each node has a single outgoing connection. (b) *Bidirectional*: each node has two outgoing connections. (c) *Unirandom*: akin to *Unidirectional*, but with an extra random outgoing connection per node.

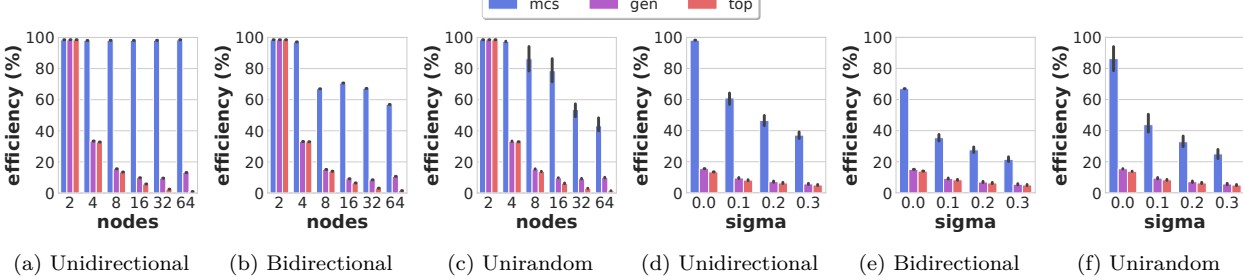

Figure 10: Efficiency comparison of *mcs* (our approach), *top*, and *gen*. In panels (a-c) the number of nodes is varied with no asynchronicity ($\sigma = 0$), while in panels (d-f) the asynchronicity levels are varied with a fixed size of $N = 8$ nodes.

# A  Scalability Analysis with Abstract Topologies

In this scalability study, we focus on artificially generated computation graphs, as they allow us to systematically vary the number of nodes, the level of asynchronicity, and the topology of the graph. We consider three different topologies: *unidirectional*, *bidirectional*, and *unirandom*, depicted in Fig. 9.

The nominal sampling time of each node is set according to the node's index $i$ as $\Delta t_i = \frac{1}{i}$ s, except for the last node's sampling time which is set to $\Delta t_N = \frac{1}{200}$ s. These topologies resemble cascaded control schemes that are common in robotic systems, with slower learning-based nodes and faster simulator nodes with intermediate controllers, estimators, sensors and actuators. The effective sampling time is computed with Eq. (2) as further detailed in Sec. 4.3.

As in Sec. 4.3, we consider a different number of nodes $N \in \{2, 4, 8, 16, 32, 64\}$, and a varying levels of asynchronicity $\sigma \in \{0, 0.1, 0.2, 0.3\}$, and replicate every experiment 5 times using different random seeds. For each configuration, we generate 20 computation graphs, each running for a duration of 100 seconds.

Figures 10a, 10b, and 10c illustrate the performance of our algorithm for different numbers of nodes when there is no asynchronicity ($\sigma = 0$). Our approach achieves a 100% efficiency for the unidirectional topology, whereas the efficiency of baseline approaches declines rapidly as the number of nodes increases. The superior efficiency of our approach in the unidirectional topology is attributed to its fewer connections. Figures 10d, 10e, and 10f demonstrate the performance of our algorithm for different levels of asynchronicity when the network comprises $N = 8$ nodes. As asynchronicity increases, partitions become more dissimilar, and the efficiency of our approach does decline, yet it remains multiples higher than that of the baseline approaches.

Figure 11 presents an analysis of our method's computational complexity in constructing the supergraph, considering both the computation graph's characteristics and topology, and the scaling of supergraph search complexity over all recorded computation graphs. Figures 11a, 11b, and 11c detail our algorithm's complexity under varying asynchronicity levels through time as it processes all recorded computation graphs. The initial episodes incur higher computational costs due to the increased computational overhead of handling numerous partial matches (Line 6-7 in Alg. 2), while subsequent episodes demonstrate linear scaling in time.

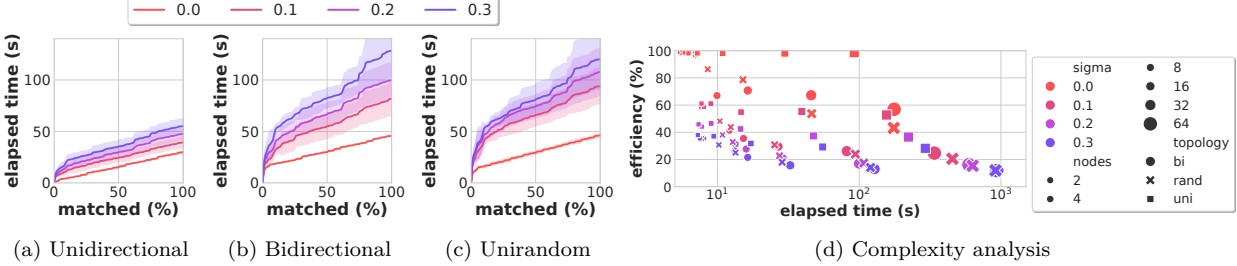

(a) Unidirectional    (b) Bidirectional    (c) Unirandom    (d) Complexity analysis

Figure 11: Performance analysis of computational complexity and efficiency. Panels (a-c) show the elapsed time for completion with $N = 32$ nodes across various asynchronicity levels and topologies. Initial episodes are time-intensive due to numerous preliminary partial matches, followed by a consistent linear time scaling in processing time. Panel (d) shows the computational complexity versus efficiency for different topologies, asynchronicity levels, and node counts, highlighting their impact on performance.

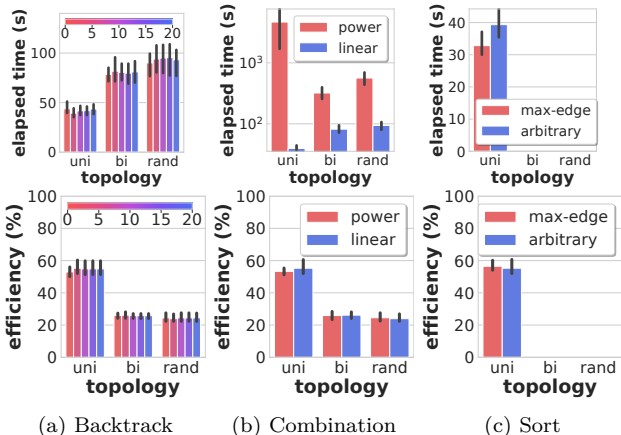

(a) Backtrack    (b) Combination    (c) Sort

Figure 12: Ablation study on topologies with $N = 32$ and $\sigma = 0.1$, examining computational complexity and efficiency. Sub-figures show: (a) Effects of varying $\beta$; (b) Efficiency-impact of considering one (*linear*) vs. all combinations per size $k$ (*power*); (c) Comparison of *arbitrary* and *max-edge* topological sorts.

In Fig. 11d, we again observe that efficiency is inversely related to both the asynchronicity level and the number of connections per node within a topology. Specifically, the unidirectional topology outperforms the bidirectional and unirandom topologies due to its fewer edges. A decrease in efficiency correlates with an increase in computation time, primarily because fewer complete matches are found, which is consistent with the results in Sec. 4.3. While the most substantial contributor to computation time is the number of nodes in the topology, it does not affect efficiency as similar efficiency is achieved with different numbers of nodes.

## B  Ablation Study

In this study, our goal is to substantiate that our approach employs simplifications discussed in Sec. 3.4 that reduce computational complexity without significantly affecting performance. For this ablation study, we focus on the topologies in Fig. 9 with $N = 32$ and $\sigma = 0.1$, ablating the proposed simplifications.

Fig. 12 illustrates that the benefits of *backtracking* are limited. However, it neither increases the computational complexity of our approach nor adversely affects efficiency.

We also analyzed the effect of considering only a single combination for each size $k$, as opposed to exploring all combinations. Fig. 12 demonstrates that this simplification has negligible impact on efficiency but considerably reduces the computational complexity (note the log-scale). It is worth noting that only considering a single combination even seems to perform slightly better in some cases. This outcome, while not statistically

implausible, may also be caused by other factors, such as the order in which the graphs are processed. Our hypothesis centers on the specific nature of the computation graphs generated by cyber-physical systems. These graphs tend to exhibit a relatively fixed structure, meaning the variety of topological orderings is considerably constrained compared to more generic graphs. Consequently, this structural rigidity could diminish the advantages we might expect from checking all combinations.

Lastly, we explored the implications of using a single topological sort. Rather than exhaustively considering all topological sorts—an approach that would be computationally prohibitive—we compared the effects of using an arbitrary sort versus a max-edge sort. The max-edge sort of the supergraph is defined as one that accommodates the maximum number of potential edges (i.e., constraints) and therefore increases the chance of finding a match in Alg. 2. Due to the inherent unidirectionality of the unidirectional topology, the max-edge sort arranges vertices of lower indices before those of higher indices. Since we lack max-edge sorting criteria for bidirectional and unirandom topologies, we limited this part of the study to the unidirectional topology. Fig. 12 shows that this simplification has negligible impact on efficiency.

# C  Graphs

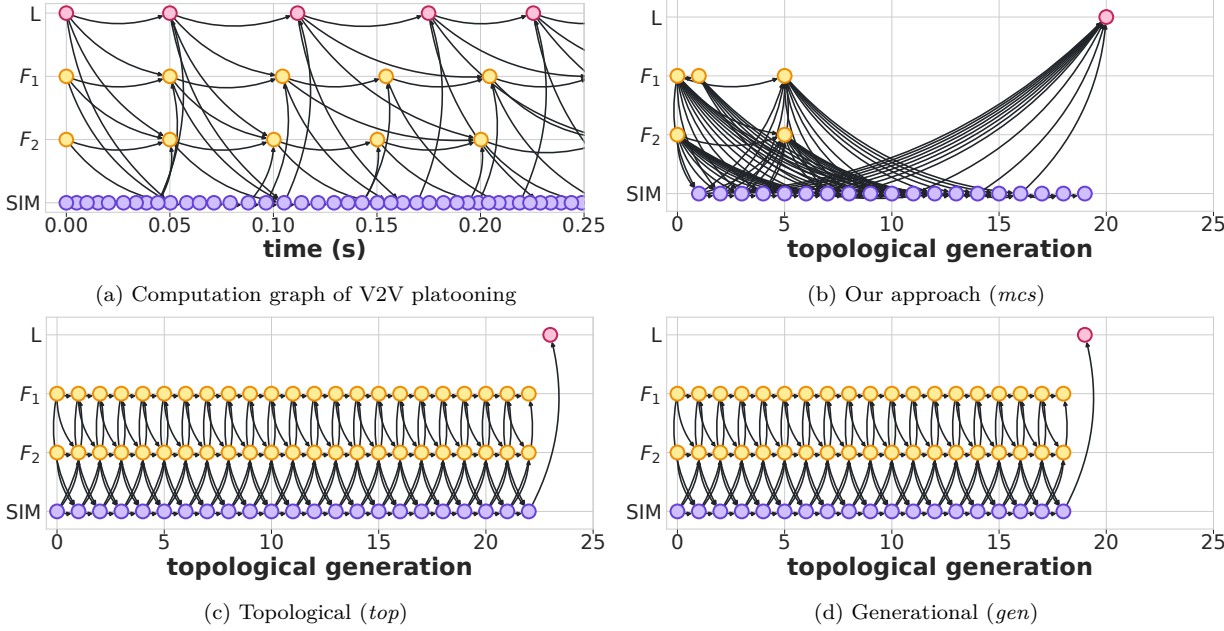

(a) Computation graph of V2V platooning

(b) Our approach ($mcs$)

(c) Topological ($top$)

(d) Generational ($gen$)

Figure 13: Panel (a) presents a segment of a computational graph corresponding to the platooning scenario in Fig. 7b with $N = 4$ and $\sigma = 0.2$. Vertices of identical color correspond to the same periodic computation unit, and edges represent data dependencies. Panels (b-d) illustrate the supergraphs generated by our method ($mcs$), as well as the topological ($top$) and generational ($gen$) methods. Our approach yields a supergraph with a reduced number of vertices, indicating enhanced efficiency in identifying commonalities across the computation graphs.

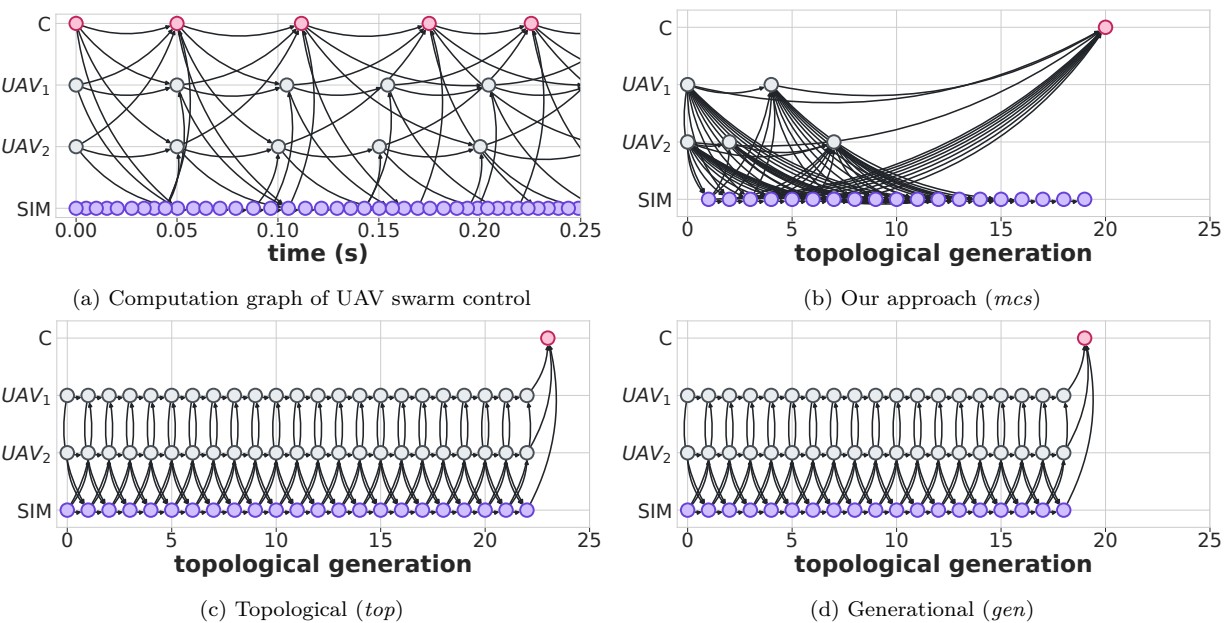

(a) Computation graph of UAV swarm control

(b) Our approach (*mcs*)

(c) Topological (*top*)

(d) Generational (*gen*)

Figure 14: Panel (a) presents a segment of a computational graph corresponding to the uav swarm scenario in Fig. 7b with $N = 4$ and $\sigma = 0.2$. Vertices of identical color correspond to the same periodic computation unit, and edges represent data dependencies. Panels (b-d) illustrate the supergraphs generated by our method (*mcs*), as well as the topological (*top*) and generational (*gen*) methods. Our approach yields a supergraph with a reduced number of vertices, indicating enhanced efficiency in identifying commonalities across the computation graphs.

