# OpenReview forum: "Efficient Parallelized Simulation of Cyber-Physical Systems"
_TMLR — Accepted by TMLR_

### Review · Reviewer_AtP5 · 2024-03-08

**Summary Of Contributions:**

This paper presents an approximate method for generating efficient accelerator computation graphs for repeated not-quite-identical simulation steps. The target domain is physics simulation, where the differences come from asynchronously operating parts with varying delays: computation, multiple sensors, actuators, physics model updates.

The paper introduces an approximate algorithm for finding the minimum size, directed acyclic supergraph that contains all model steps, from some large set of previously observed steps or synthetic data. This single supergraph can then be used to do a fast parallel (multi-environment) simulation step, with different environments delays being represented through data dependencies.

The authors provide sim-to-real experiments on two physical systems. In one of these domains, simulation speed was mostly irrelevant to real performance, but the other experiment did planning and had a real performance cost for increased simulation time. Despite the authors' delay-aware simulation steps being slower than ignoring the delays (producing simple, identical model steps that are easily and efficiently parallelisable), the proposed algorithm had better sim-to-real performance for both environments. The sequential and proposed models are compared to mock algorithms for delay-aware compuation graphs.

The authors also provide synthetic data experiments on supergraph construction time, and on the efficiency of the supergraph -- how many nodes are not masked out through data dependencies when computing steps.

**Audience:**

Yes

**Claims And Evidence:**

Yes

**Requested Changes:**

How is algorithm 2 guaranteed to only considering acyclic mappings? Is this just because S is already acyclic? How is S being guaranteed to avoid adding a cycle as it adds the different edges from different partitions?

I suspect and hope there's a fairly simple, obvious answer I'm just missing. The remaining points are all non-critical.


"Our approach addresses the minimal universal supergraph problem" -- given the acyclic constraint, would it be more accurate to say it addresses a variant of the MUG problem?

"Unlike the brute-force algorithm presented .. our greedy algorithm is" -- suggest something like "Unlike the brute-force exact algorithm presented ... our approximate greedy algorithm"

Figure 3.  Are all the lines/nodes correct?
For example, in b), how has the second magenta sensor node ended in Pi,0 when it's not an ancestor of S0?  Because i>0 and there's an existing S matching it from the first purple physics node? If so, it would be nice to have that current S at the start is this iteration i.
Why is there no edge from the third magenta (first in Pi,1) to the green node in Pi,1?  Edges from green to orange in e)?

Algorithm 1.  I'm not sure what indexing a vertex u_i means. Is that just relabeling it? Why do do this at all, rather than saying j <- index of  u: I(...) and use u_i and Pi,j Mi,j?
Later text also states "We alleviate this by considering only a single ...". This should be in the algorithm: as is, it seems to be considering all combinations and all sortings.

Section 5.1: should note here that there is a non-trivial startup cost to generate the supergraph, but this cost is only paid at startup, and is not being included in the evaluation of run-time performance.  I was glad to eventually discover that this was already noted at the end of the section on scalability, but I think it should be included at the front where it is describing how the algorithm is evaluated, either in addition to the section 5.3 comment or instead of. While I broadly agree that the startup cost can be reasonably amortised across a larger run, the slowest point in FIgure 9 is around 10^3 seconds, which makes the startup / run-time split be worth mentioning.

Figure 5: given the non-trivial difference in speed, it would be good to have the training curve with time as the x-axis -- either instead of steps, or in addition to steps if the authors would like to preserve the existing figure.

"Moreover, mcs method also results in smoother operations than seq."  Is there anything viewable to the readers to see that? If not, should soften this to something like "Moreover, in the author's subjective evaluation, the mcs method ..."

"The effective sampling time of each node is computed as delta t_i,ik  ... scaled with the nominal sampling time."  There appears to be something not quite right with this sentence. Should there be a delta t_i-1,k?  Was there previously an x_i,k that was removed? If there is a delta i_,k = delta i-1,k, it seems to be non-decreasing given the current location of max(0,...)

Ablation: the choice of "linear" (a single combination) seems to actually be better than "power" for both uni and bi. The algorithm is a greedy approximation so this is not impossible, but it's surprising.  Do the authors have an explanation?

"optimal sort": does this actually cause the algorithm to produce the optimally efficient supergraph? If not, I would suggest a different name ("max-edge sort"? something else?) that doesn't imply algorithm optimality. If it *does* imply optimal efficiency -- great! -- then fine as is.

**Strengths And Weaknesses:**

Strengths: the paper is pretty clear about the goal of enabling parallel simulation of physical environments, with an extended capability for handling delays in different components of the system. The authors make arguments for why this is worthwhile, and why the proposed method is reasonable, with some experimental justification.


Weaknesses: the motivation seems to better sim-to-real performance, resting on one fairly simple environment and a fairly small difference in absolute terms within a more complicated environment, and it's not compared to any real alternatives. Can this be strengthened? ORBIT has limited asynchronicity, but does have some delay -- how does sim-to-real performance compare if those delays are introduce in the sample environment?  Or are there are other ways to strengthen the claim that there is a significant benefit in the sim-with-delay, beyond two environments where the agent does do better (rather than lucky regularisation/bias?): how does sim-to-delay-sim compare to sim-to-real?

Note to be clear -- I do not think the current set of experiments are definitely inadequate, but I do think this might be the biggest weakness.

---

> ### Author Response · Authors · 2024-03-24
> **Response to Reviewer AtP5 (Part 1)**
>
> Dear Reviewer,
>
> Thank you for your detailed and insightful review, which stood out in its depth and attention to detail. In particular, we appreciate the detail in which you went through our manuscript that led to spotting several inconsistencies that turned out to be subtle typos. All changes have been highlighted in blue for ease of reference. Below we address your broader concerns or suggestions that require more detailed responses, and group them by topic.
>
> ===Addressing Potential Experimental Validation Weakness===
>
> In response to your concerns regarding the breadth of our experimental validation, we have included the sim-to-delay-sim experiments you proposed, to strengthen the claim that the improved sim2real performance can be attributed to the inclusion of delay simulation. Moreover, we would like to highlight that every experiment was repeated 5 times to reduce the luck factor.
>
> We do not set out to prove the importance of delay simulation—its significance is well recognized in the robotics, CPS, and control literature (e.g., [1][2][3][4][5], and [6]). Instead, our focus is on demonstrating that our novel approach, randomizing the predication mask for a supergraph, effectively bridges the sim2real gap, akin to direct delay simulation, but with added efficiency. Our choice of a simple pendulum task, a system of which we have a nearly perfect dynamical model, is intentional. It allows us to highlight the impact of delay effects in isolation. Through this, we show that our method not only emulates these delays with fidelity but also offers an efficient solution for delay simulation. Consequently, while extending our work to more complex systems could provide further insights, such an extension is not imperative for establishing the validity of the claim that our approach emulates asynchronous effects and leads to more accurate simulation.
>
> What **is** vital is demonstrating our ability to resolve the graph problem across larger system topologies. Initially, we addressed this for three abstract topologies. In the revised manuscript, we have expanded our investigation to include two real-world cyber-physical system topologies, strengthening our empirical validation for potential real-world applications.
>
> Finally, we want to emphasize that the improvements in the box pushing task are more substantial than they might appear. While a 1.2 cm increase in accuracy for a 50 cm pushing task may seem minor, achieving precision becomes increasingly challenging as the box nears its target. It's in these moments of fine manipulation, when avoiding overshoot is crucial, that the advantages of delay simulation shine, as demonstrated by a >40% improvement in final accuracy. Now, including this in an efficient manner such that it can be used in real-time control is the contribution of our work. The supplementary video highlights the smoother motion of using delay simulation relative to the baseline without simulating delays.
>
> ====Clarification on Algorithm 2's Acyclicity====
>
> We appreciate your query regarding the acyclicity of Algorithm 2, and the answer is not necessarily a simple one so it should have been addressed more explicitly in the original manuscript. In the original manuscript, this was only implicitly addressed in Section 4.3 with "By design, ... resulting graph $G_u$'" and even included a nasty typo ($P^c$ should have been $P^*$). We have now expanded on this point in the revised manuscript at the end of both Section 3.2 and 3.3.
>
> In short, this property is maintained by starting each search at the roots of $G_\text{excl}$ (Alg. 2, Line 4) and $S$, followed by a search strategy that adheres to the edge constraints of both graphs, by traversing the topological order for $S$ (Alg. 2, Line 7), and following a breadth-first search for $G_\text{excl}$ (implicitly enforced by how the search front evolves), respectively.
> In more detail: this ensures that edges connecting matched vertices within $P^*$ prevents cycle formation in the updated $S'$ when adding edges of $G_\text{excl}$, and vice versa if we were to add the edges of $S$ to $G_\text{excl}$.
> For vertices outside $P^*$ (identified as $P^-$), their position is either entirely **before** or **after** $P^*$ within the topological order of $G_\text{excl}$. This position is maintained by our selective exclusion of vertices and how we expand the search frontier:
> - Vertices can be before $P^*$ if they are excluded from the initial search frontier **if and only if** they were root vertices of $G_\text{excl}$ (Alg. 2, Line 21). Note that $F_\text{con}$ is a filtered set of root vertices as per (Alg. 2, Line 5).
> - By virtue of how the search starts from the roots of $G_\text{excl}$ and how the search frontier is updated in a breadth-first manner by only adding newly emerging roots (Alg. 2, Lines 15-16), the left-over unmatched vertices are guaranteed to be after $P^*$ in the topological order of $G_\text{excl}$.
>
> Please see Part 2 of this response!

---

> ### Author Response · Authors · 2024-03-24
> **Response to Reviewer AtP5 (Part 2)**
>
> ====Other Changes and Comments====
>
> We've implemented the suggested changes in the manuscript, and for some, provided additional motivation for our choices:
>
> In the updated manuscript, we now properly mention the initial cost of generating the supergraph. However, we chose not to alter the x-axis of the training curve in Figure 5c, asserting that a speed evaluation should omit the learning algorithm's overhead. This perspective aligns with the findings in the Brax paper in Section 5.1.2 [7], where they noted SAC's superior sample efficiency compared to PPO and identified training throughput limitations primarily due to SGD updates (12% environment execution, 10% replay buffer operations, 78% SGD updates). Our stance is that assessing speed based solely on environment execution provides a clearer picture, excluding learning algorithm-related overhead. We have appended a note in the evaluation section to elucidate this approach. Consequently, readers can more straightforwardly gauge the potential benefits of applying our method across various algorithms (e.g., OpenES, PPO) without needing to discount SAC's specific overhead influences.
>
> To transparently indicate the points at which approximations are applied within our algorithm, and due to space limitations in the manuscript, we opted not to replicate the entire algorithm. Instead, we introduced a direct annotation within the algorithm's text. By incorporating comments at specific lines where approximations occur, we provide a straightforward method for readers to identify these approximations in the implemented algorithm. This approach ensures clarity and precision without necessitating extensive additional text or diagrams.
>
> We were initially surprised to observe that the "linear" approach outperformed the "power" approach, which, as you correctly noted, should theoretically perform equally well or better given that the linear approach's possibilities are a subset within the power approach. This outcome, while not statistically implausible, may also be caused by other factors, such as the order in which the graphs are processed. Our guess centers on the specific nature of the computation graphs generated by cyber-physical systems. These graphs tend to exhibit a relatively fixed structure, meaning the variety of topological orderings is considerably constrained compared to more generic graphs. Consequently, this structural rigidity could diminish the advantages we might expect from the power approach due to the limited scope for graph combination. We have included a remark in the manuscript's revised version to address and reflect on this unanticipated finding.
>
> Thank you for your valuable insights that have helped refine our paper. We appreciate your guidance and are ready to address any more suggestions or clarifications you might suggest.
>
> - [1] Leader tracking in homogeneous vehicle platoons with broadcast delays. (Peters et al., 2014)
> - [2] Learning-based time delay attack characterization for cyber-physical systems. (Lou et al, 2019)
> - [3] Distributed model predictive longitudinal control for a connected autonomous vehicle platoon with dynamic information flow topology. (Zhao et al., 2021)
> - [4] A survey on platoon-based vehicular cyber-physical systems. (Jia et al., 2015)
> - [5] Drone swarms as networked control systems by integration of networking and computing. (Asaamoning et al, 2021)
> - [6] A hybrid approach to cyber-physical systems verification. (Kumar et al., 2012)
> - [7] Brax -- A Differentiable Physics Engine for Large Scale Rigid Body Simulation. (C. Daniel Freeman et al. 2021)

---

> > ### Comment · Reviewer_AtP5 · 2024-03-25
> >
> > Thanks to the authors for making the set of changes, including an entirely new set of experiments. The in-mcs-environment results in the pendulum environment are what one would hope to see, and the multi-vehicle environments are a decent step away from the simplicity of a pendulum environment.
> >
> > I would consider my comments and questions addressed.

---

### Review · Reviewer_Arj5 · 2024-03-13

**Summary Of Contributions:**

This paper focuses on the challenge of accurately simulating the asynchronous and delay-filled interactions that occur in cyber-physical systems. While the advancements in accelerated physics simulations have significantly reduced the training times for reinforcement learning policies, these traditional simulations fail to capture the real-time, asynchronous nature of real-world interactions, leading to a notable gap between simulated and real-world performance (the "sim2real gap"). The paper attempts to develop an efficient parallel approach for cyber-physical system simulations on accelerator hardware, including physics, sensors, actuators, processing components and their asynchronous interactions.

**Audience:**

Yes

**Broader Impact Concerns:**

No broader impact concerns.

**Claims And Evidence:**

Yes

**Requested Changes:**

Overall, the paper presents a promising approach to bridging the sim2real gap in cyber-physical systems by accurately simulating asynchronicity and delays. But the evaluation of the approach is very limited and the paper is not yet ready for publication. Addressing the weaknesses above would make the paper stronger.

Scaling to CPS systems such as an autonomous car, an UAV, a robot, etc will require addressing a lot of challenges left unaddressed in the paper. For e.g., different observed variables in CPS evolve at different time-scales. At the very least, one can consider a parametric system (e.g, traffic collision avoidance TCAS system with increasing number of aircrafts) and study how the scalability changes with the parameter. The reviewer would request authors to look at academic (and small to moderate) sized CPS benchmarks at https://github.com/lyudeyun/AI-CPS-Benchmark and https://github.com/natchi92/BASBenchmarks

The new experiments address the primary concern of this reviewer.

**Strengths And Weaknesses:**

Strengths:

- The paper's approach addresses the key issue of accurately simulating the asynchronous interactions between computational elements, sensors, and actuators in cyber-physical systems. This makes the simulator more reflective of actual conditions.

- The construction of supergraph that encodes all data dependencies across parallelized simulation steps, ensuring that simulation steps are processed in a sequence that respects these dependencies without violating them, is very interesting.

- The goal of finding the smallest supergraph possible and sorting it topologically, minimizes redundant computation. This efficiency is critical in maintaining high simulation speeds and reducing the needed computational resources.

Weaknesses/Questions

- Identifying the smallest supergraph is a complex, NP-hard problem, which could potentially limit the approach's applicability in scenarios where computational resources are constrained or when simulating exceedingly complex systems. It would be useful to relate the assumptions used in the greedy approximation to real-world constraints on a CPS. For example, isn't it normal for different nodes to be stateful?

- The evaluation is over two very simple systems - pendulum and box pushing manipulator. Given the focus on large scale CPS system simulation, these examples are not sufficient to demonstrate practical utility.

- The relevance to machine learning is tenuous. Demonstrating how this efficient simulation can be helpful for some learning task would be useful.

---

> ### Author Response · Authors · 2024-03-24
> **Response to Reviewer Arj5**
>
> Dear Reviewer,
>
> Thank you for your time and effort in reviewing our manuscript. We have made the necessary changes to the manuscript, which are highlighted in blue for ease of reference. Below, we address your concerns and suggestions that require more detailed responses, and group them by topic.
>
> ===Assumptions & Approximations===
>
> We appreciate the emphasis on the necessity to clarify the assumptions and approximations inherent in our approach. Notice, however,  that we already addressed these points in the original manuscript, specifically in the section titled "Limitations and Approximations." In this section, we acknowledge the stateful nature of components, as mentioned, and elaborate on with the statement, "These assumptions are particularly well-suited for cyber-physical systems where components are stateful and run at fixed frequencies...."
>
> ====Demonstration in Learning Task====
>
> To clarify, our manuscript already includes a detailed demonstration of our approach's effectiveness in two learning tasks (pendulum, box pushing), underscoring its potential to enhance sim2real performance. Specifically, we presented evidence showing how integrating our approach with two established ML methods, SAC [1] and CEM [2]—both of which have been published at machine learning conferences (ICML, CoRL)—results in superior performance on two real-world tasks. Therefore, we believe our manuscript already demonstrates the potential of our approach in a learning task. However, we may be misunderstanding your point, and we are open to further clarification to ensure our response aligns with your expectations. Finally, we would like to direct attention to our response to Reviewer DfLD, where we discuss the relevance of our method to machine learning.
>
> ===Large scale CPS===
>
> Notice that our initial manuscript did include a scalability analysis for larger systems in Section 5.3.  Moreover, before we were notified of your updated review from OpenReview, we had already expanded our evaluation to include two additional real-world cyber-physical systems: V2V Platooning [3] and UAV swarm control [4]. We believe these systems align with the spirit of your suggestions and hope that they address your concerns. We also invite your attention to the original scalability analysis now detailed in Appendix A. This analysis illustrates the ability of our approach to efficiently handle large-scale systems of up to 64 unique nodes with diverse time-scales and delay levels, resulting in supergraphs comprising over 2000 vertices. Our findings showed a tenfold efficiency improvement over baseline methods, with efficiency gains, in fact, increasing compared to the baselines when the system's size and complexity increased.
>
> Our primary aim in the "Performance" Section is not to reiterate the well-recognized value of delay simulation (e.g., [1][2][3][4][5], and [6])  but to validate our method—randomizing the predication mask for a given supergraph—as an efficient alternative to direct delay simulation. We demonstrate this using smaller systems, such as the deliberately chosen simplified pendulum task, which offers a clear, isolated view of delay effects, underscoring our method's capability in addressing the sim2real gap efficiently.
>
> We concur that the two real systems tested in our laboratory may not be classified as large scale, yet it is important to emphasize that they are indeed cyber-physical systems characterized by multiple time scales where accurately simulating delays is crucial. Consequently, while extending our work to more complex systems could provide further insights, such an extension is not imperative for establishing the validity of the claim that our approach emulates asynchronous effects and leads to more accurate simulation.
>
> We believe that the scalability experiments and the inclusion of two additional real-world topologies demonstrates our approach's scalability and that the real-world systems  validate the accuracy claims. In all, your insights have been instrumental in refining our manuscript and remain open to any further suggestions or clarifications you deem necessary.
>
> - [1] Soft actor-critic: Off-policy maximum entropy deep reinforcement learning with a stochastic actor. (Haarnoja et al., 2018)
> - [2] Data efficient reinforcement learning for legged robots. (Yang et al., 2020)
> - [3] Distributed model predictive longitudinal control for a connected autonomous vehicle platoon with dynamic information flow topology. (Zhao et al., 2021)
> - [4] Drone swarms as networked control systems by integration of networking and computing. (Asaamoning et al, 2021)
> - [5] Leader tracking in homogeneous vehicle platoons with broadcast delays. (Peters et al., 2014)
> - [6] Learning-based time delay attack characterization for cyber-physical systems. (Lou et al, 2019)
> - [7] A survey on platoon-based vehicular cyber-physical systems. (Jia et al., 2015)
> - [8] A hybrid approach to cyber-physical systems verification. (Kumar et al., 2012)

---

> > ### Comment · Reviewer_Arj5 · 2024-03-26
> > **Thank you**
> >
> > Thanks for including the additional evaluations. This addresses the primary concern of the reviewer.

---

### Review · Reviewer_DfLD · 2024-03-14

**Summary Of Contributions:**

This paper presents a technique for efficiently simulating asynchronous physical systems. When simulations are asynchronous, there is a nontrivial computation graph that arises where individual environment steps can no longer be parallelized on a GPU. This paper presents a novel method for finding a partitioning strategy of the multi-step asynchronous computation graph that allows for more effective parallelization on a GPU. The paper then shows that this asynchronicity matters for sim2real transfer when training an RL agent. By matching the asynchronicity of a real system in simulation, the RL agent trained in simulation better transfers to the real world. Similarly, the paper shows that the presented method for finding static computation graphs leads to more efficient simulation than baseline approaches.

**Audience:**

Yes

**Claims And Evidence:**

Yes

**Requested Changes:**

* Presentation clarity needs to be improved for a machine learning audience
* Examples
  * Figure 2 could use work in a few ways
    * what is the difference between the red and black arrows?
    * what do the different node colors mean? Do they correspond to Figure 1? Could these be labelled here again?
    * why is the supervisor node always the top green one? does it need to be?
  * Predication mask is not clearly defined in the text -- what is it? Control flow to data flow dependencies was not super clear.
  * Can the authors give some examples of computation graphs that would come up for a given agent interacting with a given environment? This would help ground several of the algorithms into meaningful examples that might be easier to parse for a machine learning audience
  * In general, it would be helpful to refer to the different types of nodes with names that indicate their physical system counterparts, as done in Figure 1
  * In the experimental section on computation graph topologies, it would similarly be useful to give grounded physical simulation examples of when each of those topologies would come up, and why they would matter.
  * Many terms in the related work section on minimum common supergraphs are undefined, and will be unknown to a machine learning audience. It might make more sense to move this section later in the paper, after preliminaries and the details of the algorithm. (e.g. what a monomorphism is, what "post-merge" means, etc.)

* Questions about experiments
  * Is the time for algorithm 1 / 2 to be run included in the overall timing comparisons between MCS and seq? Could this be clarified in the paper?

Smaller changes
* be consistent with capitalization, use MCS always or mcs always, but do not mix between the two
* in the related work section: “Furthermore, we discover the set of graphs alongside the mcs, whereas Trimble (2021) and Bunke et al. (2003) presume a given graph set.”
  * Earlier it was stated that the minimal universal supergraph problem … “ seeks the smallest supergraph, i.e., the minimum common supergraph (mcs), containing all graphs in a given set as a subgraph (Trimble, 2021)”
  * This seems contradictory to the statement above then, as the graph set has to be initially given according to the problem statement. Could this be explained?
* Notation could be improved
  * |G| denotes the number of vertices in G, but does not include edges. Consider changing to |V| which is more direct.
  * "The edges that connect G1 and G2 − G1 are defined as the embedding embG2 (G1), which is a subset of E(G2)"
    * this will be confusing for a machine learning audience, as embeddings typically mean the outputs of a neural network.

**Strengths And Weaknesses:**

Strengths
* Highlighting an interesting and overlooked problem in sim2real transfer.
  * The paper highlights the challenges involved in dealing with real systems, which are often far from their idealized counterparts in simulation. The kinds of asynchronous timings that the paper discusses are common, and existing commonly used simulators often do not account for them. The experiments run in the paper highlight that this is a problem for sim2real transfer, and show that the presented asynchronous simulator supports better transfer of RL policies to the real world.
* Novel greedy algorithm for discovering Minimum Common Supergraphs
  * As this is not my area of expertise, I cannot directly comment on the novelty of this contribution, but the paper claims that their greedy approach to finding a minimum common supergraph is novel, and it appears to work better than two reasonable sounding baselines. This is not a machine learning contribution, however.

Weaknesses
* Relevance to machine learning audience
  * By the paper's own statement: "The main contribution of this paper is an approach to parallelize cyber-physical system simulations that
emulates asynchronicity and delays with minimized computational overhead on accelerator hardware." This is very interesting, but is not really about machine learning. The main contribution instead relates to software optimization. The main innovations relating to finding super graphs do not involve any machine learning.

  * Instead, the way in which this is connected to machine learning is the fact that the sim2real performance is tested with a reinforcement learning agent. However, this is a somewhat tenuous connection to machine learning. Without source code for the GPU optimization being released with the paper (instead only Algorithms 1 and 2 are provided), it is hard to imagine that TMLR's audience members would use these algorithms for GPU optimization, meaning that it would likely not be adopted.

* Presentation and clarity of writing
  * The paper is similarly written with a software optimization audience in mind. As a result, it will be difficult for readers of TMLR to understand. There are many undefined terms throughout the paper (for example, what is a predication mask? monomorphism? What is control flow vs. data flow?) which might be obvious to people with a software optimization background, but which are less likely to be familiar to readers coming from a machine learning background.

---

> ### Author Response · Authors · 2024-03-24
> **Response to Reviewer DfLD**
>
> Dear Reviewer,
>
> Thank you for the detailed feedback on our manuscript. We have carefully considered your comments and made the requested changes to the manuscript. All changes have been highlighted in blue for ease of reference. Below, we address your concerns and suggestions that require more detailed responses, and group them by topic.
>
> ====Relevance to ML Audience====
>
> We  believe that our work does hold substantial relevance for the ML community, drawing parallels with the contributions made by Brax [1] and Nvidia Isaac Gym [2], that were both presented at NeurIPS 2021. These studies have been instrumental in advancing reinforcement learning, demonstrating the field's interest in such innovations.
> Given that dynamical systems frequently operate within the inner loop of ML algorithms, optimizing their efficiency and adaptability to accelerated hardware is crucial.
> Any improvement in this area directly enhances the performance of learning algorithms, underlining the importance and relevance of our study to the field.
>
> While Brax [1] and Nvidia Isaac Gym [2] concentrated on speeding up physics simulation, our research targets the acceleration of dynamics influenced by delays within graph-like networks, potentially incorporating elements of Brax [1] or Nvidia Isaac Gym [2] as vertices. Indeed, the success of Brax and Nvidia Isaac Gym could be attributed to their user-friendly, plug-and-play solutions, which don't necessitate ML practitioners to possess extensive knowledge of the physics involved. If your recommendation is that we should emulate this approach by offering a straightforward tool in the same spirit, then we believe that is an excellent suggestion. This strategy would likely increase our work's utility within the ML community.
> Therefore, we will include a user-friendly version of our compiler in the supplementary material, featuring our supergraph method at its core. This tool will enable users to enhance dynamical systems, modeled as networks of nodes, with efficient delay simulation capabilities. Essentially, it accepts computation graphs and compiles them into an environment with a gym-like interface, complete with 'reset' and 'step' functions, facilitating its integration as a seamless replacement for the original environment. Instructions and an example usage of this compiler will be provided in the supplementary material to assist users in leveraging our approach effectively. To facilitate immediate discussion on other aspects of the paper, we have chosen to submit our revised version and responses to other concerns now. We will ensure the compiler is provided by March 28th for review before your final decision.
>
> Finally, ML algorithms, in general, may result in dynamic computation graphs and pose challenges for libraries like Jax and TensorFlow, which use static graphs. This issue was explored in previous research [3] presented at ICLR. Similarly, our approach, by aligning dynamic computation graphs with a static supergraph, holds promise for enhancing the integration of machine learning algorithms that generate dynamic graphs into frameworks like Jax and TensorFlow.  We have included a brief discussion of this potential in the Conclusion of the revised manuscript.
>
> ====Presentation and Clarity for ML Audience====
>
> We have revised the manuscript to make it more accessible to a broader ML audience. Any additional feedback on the clarity of the revised manuscript would be greatly appreciated. Although the core graph-theoretical elements of our algorithm are complex, we've made the explanation in Section 3 clearer for ML readers, focusing on the problem and our solution. We've also updated Figure 2 to be more intuitive based on your suggestions. Finally, the positioning of the vertices in the figures is purely for enhancing visualization clarity. Ultimately, it is the relationships between vertices and their edges that are crucial.
>
> Per your suggestion, we have added topological experiments with two relatable real-world examples, also in line with the feedback by Reviewer Arj5. The abstract topologies are still included in Appendix A. Appendix C provides visualizations of example computation graphs together with the identified supergraphs of our method, compared with baseline methods.
>
> We have moved the related works section to the end of the manuscript, and clarified the seemingly contradictory statement and confusion of using MCS vs mcs.
>
> Thank you for your feedback, and we welcome any additional comments or questions you may have.
> - [1] Brax -- A Differentiable Physics Engine for Large Scale Rigid Body Simulation. (C. Daniel Freeman et al. 2021, Neurips)
> - [2] Isaac Gym: High Performance GPU-Based Physics Simulation For Robot Learning. (Makoviychuk et al., 2021, Neurips)
> - [3] Deep Learning with Dynamic Computation Graphs. (Looks et al., 2017, ICLR)
> - [4] Graph clustering using the weighted minimum common supergraph. (Bunke, Horst, et al. 2003)

---

> ### Author Response · Authors · 2024-03-28
> **Compiler notebook now available in the supplementary material!**
>
> We would like to inform you that the compiler has been integrated into the Supergraph package, which is now included in the supplementary materials. Additionally, we've added a notebook that guides users through setting up and running simulations with the Supergraph package. It focuses on the disk pendulum system, utilized in our experiments, to demonstrate the creation of custom nodes using Brax, simulation of dynamics, and visualization of outcomes. While the notebook demonstrates the core functionality, we plan to offer an extensive and more general guide on GitHub. To maintain anonymity during the review process, access to this repository will be granted post-review.  The notebook has been tested on Google Colab, which may best way for viewing and interacting.
>
> Thank you for your suggestion! It's really helped us make our work more accessible and useful. Please feel free to offer any additional feedback that might enhance our paper or supplementary material.

---

> ### Comment · Reviewer_DfLD · 2024-04-06
> **Thank you for the clarifications and additional experiments**
>
> My questions and concerns have been addressed, thank you for the significant improvements to the paper!

---

### Decision · Action_Editor_jWed · 2024-04-17

**Recommendation:** Accept with minor revision

**Comment:**

As noted all three reviewers voted to accept and are happy with the paper.

The AE has two requests:
1. There are whitespace formatting issues to address
2. The intro should include a clear definition of CPSs and a description of one application. Both are implicitly included in the paper (the definition is a sub-clause of a sentence), but making these more direct and intentional would improve the accessibility and potential impact of the paper.

**Audience:**

This paper focuses on simulating Cyber-Physical Systems---deployment scenarios where distributed hardware and software interact, like smart grid or multi-vehicle applications. In particular this work focuses the sim2real gap attempting to capture real-time asynchronous nature of real-world interaction in Cyber-Physical Systems. Clearly an important area of study.

**Claims And Evidence:**

The paper clearly meets the bar on claims matching evidence. This aspect was improved via author-reviewer interaction where the claims were sharpened and new experiments added. To quote one of the reviewers:

"The authors did a reasonable job describing what problems they are and are not solving, and the paper was clear enough that I think I could implement the algorithms. The experiments are enough to suggest to me that it's worth trying.There is still some question of whether things scale to very large problems...". In the AE's view almost no methods in ML would pass this bar and "very large" problems are hard to define. The point on scaling is highlighted here simply because it was a concern of multiple reviewers, but in the end all three were satisfied and voted to accept.